# Beyond temperature scaling: Obtaining well-calibrated multiclass probabilities with Dirichlet calibration

**Meelis Kull**
Department of Computer Science
University of Tartu
meelis.kull@ut.ee

**Miquel Perello-Nieto**
Department of Computer Science
University of Bristol
miquel.perellonieto@bris.ac.uk

**Markus Kängsepp**
Department of Computer Science
University of Tartu
markus.kangsepp@ut.ee

**Telmo Silva Filho**
Department of Statistics
Universidade Federal da Paraíba
telmo@de.ufpb.br

**Hao Song**
Department of Computer Science
University of Bristol
hao.song@bristol.ac.uk

**Peter Flach**
Department of Computer Science
University of Bristol and
The Alan Turing Institute
peter.flach@bristol.ac.uk

## Abstract

Class probabilities predicted by most multiclass classifiers are uncalibrated, often tending towards over-confidence. With neural networks, calibration can be improved by temperature scaling, a method to learn a single corrective multiplicative factor for inputs to the last softmax layer. On non-neural models the existing methods apply binary calibration in a pairwise or one-vs-rest fashion. We propose a natively multiclass calibration method applicable to classifiers from any model class, derived from Dirichlet distributions and generalising the beta calibration method from binary classification. It is easily implemented with neural nets since it is equivalent to log-transforming the uncalibrated probabilities, followed by one linear layer and softmax. Experiments demonstrate improved probabilistic predictions according to multiple measures (confidence-ECE, classwise-ECE, log-loss, Brier score) across a wide range of datasets and classifiers. Parameters of the learned Dirichlet calibration map provide insights to the biases in the uncalibrated model.

## 1 Introduction

A probabilistic classifier is *well-calibrated* if among test instances receiving a predicted probability vector $p$, the class distribution is (approximately) distributed as $p$. This property is of fundamental importance when using a classifier for cost-sensitive classification, for human decision making, or within an autonomous system. Due to overfitting, most machine learning algorithms produce over-confident models, unless dedicated procedures are applied, such as Laplace smoothing in decision trees [8]. The goal of *(post-hoc) calibration methods* is to use hold-out validation data to learn a *calibration map* that transforms the model's predictions to be better calibrated. Meteorologists were among the first to think about calibration, with [3] introducing an evaluation measure for probabilistic forecasts, which we now call Brier score; [21] proposing reliability diagrams, which allow us

to visualise calibration (reliability) errors; and [6] discussing proper scoring rules for forecaster evaluation and the decomposition of these loss measures into calibration and refinement losses. Calibration methods for binary classifiers have been well studied and include: logistic calibration, also known as 'Platt scaling' [24]; binning calibration [26] with either equal-width or equal-frequency bins; isotonic calibration [27]; and beta calibration [15]. Extensions of the above approaches include: [22] which performs Bayesian averaging of multiple calibration maps obtained with equal-frequency binning; [23] which uses near-isotonic regression to allow for some non-monotonic segments in the calibration maps; and [1] which introduces a non-parametric Bayesian isotonic calibration method.

Calibration in multiclass scenarios has been approached by decomposing the problem into $k$ one-vs-rest binary calibration tasks [27], one for each class. The predictions of these $k$ calibration models form unnormalised probability vectors, which, after normalisation, are not guaranteed to be calibrated. Native multiclass calibration methods were introduced recently with a focus on neural networks, including: matrix scaling, vector scaling and temperature scaling [9], which can all be seen as multiclass extensions of Platt scaling and have been proposed as a calibration layer which should be applied to the logits of a neural network, before the softmax layer. An alternative to *post-hoc* calibration is to modify the classifier learning algorithm itself: MMCE [17] trains neural networks by optimising the combination of log-loss with a kernel-based measure of calibration loss; SWAG [19] models the posterior distribution over the weights of the neural network and then samples from this distribution to perform Bayesian model averaging; [20] proposed a method to transform the classification task into regression and to learn a Gaussian Process model. Calibration methods have been proposed for the regression task as well, including a method by [13] which adopts isotonic regression to calibrate the predicted quantiles. The theory of calibration functions and empirical calibration evaluation in classification was studied by [25], also proposing a statistical test of calibration.

While there are several calibration methods tailored for deep neural networks, we propose a general-purpose, natively multiclass calibration method called *Dirichlet calibration*, applicable for calibrating any probabilistic classifier. We also demonstrate that the multiclass setting introduces numerous subtleties that have not always been recognised or correctly dealt with by other authors. For example, some authors use the weaker notion of *confidence calibration* (our term), which requires only that the classifier's predicted probability for what it considers the most likely class is calibrated. There are also variations in the evaluation metric used and in the way calibrated probabilities are visualised. Consequently, Section 2 is concerned with clarifying such fundamental issues. We then propose the approach of Dirichlet calibration in Section 3, present and discuss experimental results in Section 4, and conclude in Section 5.

## 2   Evaluation of calibration and temperature scaling

Consider a probabilistic classifier $\hat{\mathbf{p}} : \mathcal{X} \to \Delta_k$ that outputs class probabilities for $k$ classes $1, \ldots, k$. For any given instance $\mathbf{x}$ in the feature space $\mathcal{X}$ it would output some probability vector $\hat{\mathbf{p}}(\mathbf{x}) = (\hat{p}_1(\mathbf{x}), \ldots, \hat{p}_k(\mathbf{x}))$ belonging to $\Delta_k = \{(q_1, \ldots, q_k) \in [0,1]^k \mid \sum_{i=1}^{k} q_i = 1\}$ which is the $(k-1)$-dimensional probability simplex over $k$ classes.

**Definition 1.** *A probabilistic classifier* $\hat{\mathbf{p}} : \mathcal{X} \to \Delta_k$ *is* **multiclass-calibrated***, or simply* **calibrated***, if for any prediction vector* $\mathbf{q} = (q_1, \ldots, q_k) \in \Delta_k$*, the proportions of classes among all possible instances* $\mathbf{x}$ *getting the same prediction* $\hat{\mathbf{p}}(\mathbf{x}) = \mathbf{q}$ *are equal to the prediction vector* $\mathbf{q}$*:*

$$P(Y = i \mid \hat{\mathbf{p}}(X) = \mathbf{q}) = q_i \qquad for \ i = 1, \ldots, k. \tag{1}$$

One can define several weaker notions of calibration [25] which provide necessary conditions for the model to be fully calibrated. One of these weaker notions was originally proposed by [27], requiring that all one-vs-rest probability estimators obtained from the original multiclass model are calibrated.

**Definition 2.** *A probabilistic classifier* $\hat{\mathbf{p}} : \mathcal{X} \to \Delta_k$ *is* **classwise-calibrated***, if for any class $i$ and any predicted probability $q_i$ for this class:*

$$P(Y = i \mid \hat{p}_i(X) = q_i) = q_i. \tag{2}$$

Another weaker notion of calibration was used by [9], requiring that among all instances where the probability of the most likely class is predicted to be $c$ (the *confidence*), the expected accuracy is $c$.

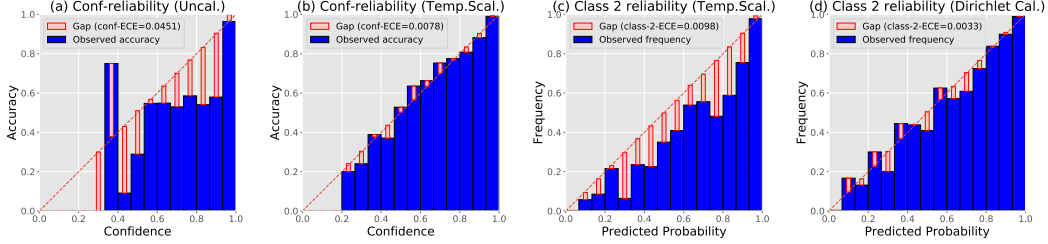

Figure 1: Reliability diagrams of `c10_resnet_wide32` on CIFAR-10: (a) confidence-reliability before calibration; (b) confidence-reliability after temperature scaling; (c) classwise-reliability for class 2 after temperature scaling; (d) classwise-reliability for class 2 after Dirichlet calibration.

**Definition 3.** *A probabilistic classifier* $\hat{\mathbf{p}} : \mathscr{X} \to \Delta_k$ *is* **confidence-calibrated**, *if for any* $c \in [0,1]$:

$$P\Big(Y = argmax\big(\hat{\mathbf{p}}(X)\big) \,\Big|\, max\big(\hat{\mathbf{p}}(X)\big) = c\Big) = c. \tag{3}$$

For practical evaluation purposes these idealistic definitions need to be relaxed. A common approach for checking confidence-calibration is to do equal-width binning of predictions according to confidence level and check if Eq.(3) is approximately satisfied within each bin. This can be visualised using the *reliability diagram* (which we will call the **confidence-reliability diagram**), see Fig. 1a, where the wide blue bars show observed accuracy within each bin (empirical version of the conditional probability in Eq.(3)), and narrow red bars show the gap between the two sides of Eq.(3). With accuracy below the average confidence in most bins, this figure about a wide ResNet trained on CIFAR-10 shows over-confidence, typical for neural networks which predict probabilities through the last softmax layer and are trained by minimising cross-entropy.

The calibration method called **temperature scaling** was proposed by [9] and it uses a hold-out validation set to learn a single temperature-parameter $t > 0$ which decreases confidence (if $t > 1$) or increases confidence (if $t < 1$). This is achieved by rescaling the logit vector $\mathbf{z}$ (input to softmax $\sigma$), so that instead of $\sigma(\mathbf{z})$ the predicted class probabilities will be obtained by $\sigma(\mathbf{z}/t)$. The confidence-reliability diagram in Fig. 1b shows that the same `c10_resnet_wide32` model has come closer to being confidence-calibrated after temperature scaling, having smaller gaps to the accuracy-equals-confidence diagonal. This is reflected in a lower *Expected Calibration Error* (**confidence-ECE**), defined as the average gap across bins, weighted by the number of instances in the bin. In fact, confidence-ECE is low enough that the statistical test proposed by [25] with significance level $\alpha = 0.01$ does not reject the hypothesis that the model is confidence-calibrated (p-value 0.017). The main idea behind this test is that for a perfectly calibrated model, ECE against actual labels is in expectation equal to the ECE against pseudo-labels which have been drawn from the categorical distributions corresponding to the predicted class probability vectors. The above p-value was obtained by randomly drawing 10,000 sets of pseudo-labels and finding 170 of these to have higher ECE than the actual one.

While the above temperature-scaled model is (nearly) confidence-calibrated, it is far from being classwise-calibrated. This becomes evident in Fig 1c, demonstrating that it systematically overestimates the probability of instances to belong to class 2, with predicted probability (x-axis) smaller than the observed frequency of class 2 (y-axis) in all the equal-width bins. In contrast, the model systematically under-estimates class 4 probability (Supplementary Fig. 12a). Having only a single tuneable parameter, temperature scaling cannot learn to act differently on different classes. We propose plots such as Fig. 1c,d across all classes to be used for evaluating classwise-calibration, and we will call these the **classwise-reliability diagrams**. We propose **classwise-ECE** as a measure of classwise-calibration, defined as the average gap across all classwise-reliability diagrams, weighted by the number of instances in each bin:

$$\text{classwise} - \text{ECE} = \frac{1}{k} \sum_{j=1}^{k} \sum_{i=1}^{m} \frac{|B_{i,j}|}{n} |y_j(B_{i,j}) - \hat{p}_j(B_{i,j})| \tag{4}$$

where $k, m, n$ are the numbers of classes, bins and instances, respectively, $|B_{i,j}|$ denotes the size of the bin, and $\hat{p}_j(B_{i,j})$ and $y_j(B_{i,j})$ denote the average prediction of class $j$ probability and the actual

proportion of class $j$ in the bin $B_{i,j}$. The contribution of a single class $j$ to the classwise-ECE will be called **class-$j$-ECE**. As seen in Fig. 1(d), the same model gets closer to being *class-2-calibrated* after applying our proposed Dirichlet calibration. By averaging class-$j$-ECE across all classes we get the overall classwise-ECE which for temperature scaling is $cwECE = 0.1857$ and for Dirichlet calibration $cwECE = 0.1795$. This small difference in classwise-ECE appears more substantial when running the statistical test of [25], rejecting the null hypothesis that temperature scaling is classwise-calibrated ($p < 0.0001$), while for Dirichlet calibration the decision depends on the significance level ($p = 0.016$). A similar measure of classwise-calibration called $L^2$ *marginal calibration error* was proposed in a concurrent work by [16].

Before explaining the Dirichlet calibration method, let us highlight the fundamental limitation of evaluation using any of the above reliability diagrams and ECE measures. Namely, it is easy to obtain almost perfectly calibrated probabilities by predicting the overall class distribution, regardless of the given instance. Therefore, it is always important to consider other evaluation measures as well. In addition to the error rate, the obvious candidates are proper losses (such as Brier score or log-loss), as they evaluate probabilistic predictions and decompose into calibration loss and refinement loss [14]. Proper losses are often used as objective functions in post-hoc calibration methods, which take an uncalibrated probabilistic classifier $\hat{\mathbf{p}}$ and use a hold-out validation dataset to learn a calibration map $\hat{\mu} : \Delta_k \to \Delta_k$ that can be applied as $\hat{\mu}(\hat{\mathbf{p}}(\mathbf{x}))$ on top of the uncalibrated outputs of the classifier to make them better calibrated. Every proper loss is minimised by the same calibration map, known as the *canonical calibration function* [25] of $\hat{\mathbf{p}}$, defined as

$$\mu(\mathbf{q}) = (P(Y = 1 \mid \hat{\mathbf{p}}(X) = \mathbf{q}), \ldots, P(Y = k \mid \hat{\mathbf{p}}(X) = \mathbf{q}))$$

The goal of Dirichlet calibration, as of any other post-hoc calibration method, is to estimate this canonical calibration map $\mu$ for a given probabilistic classifier $\hat{\mathbf{p}}$.

## 3  Dirichlet calibration

A key decision in designing a calibration method is the choice of parametric family. Our choice was based on the following desiderata: (1) the family needs enough capacity to express biases of particular classes or pairs of classes; (2) the family must contain the identity map for the case where the model is already calibrated; (3) for every map in the family we must be able to provide a semi-reasonable synthetic example where it is the canonical calibration function; (4) the parameters should be interpretable to some extent at least.

**Dirichlet calibration map family.** Inspired by beta calibration for binary classifiers [15], we consider the distribution of prediction vectors $\hat{\mathbf{p}}(\mathbf{x})$ separately on instances of each class, and assume these $k$ distributions are Dirichlet distributions with different parameters:

$$\hat{\mathbf{p}}(X) \mid Y = j \sim \mathrm{Dir}(\alpha^{(j)}) \tag{5}$$

where $\alpha^{(j)} = (\alpha_1^{(j)}, \ldots, \alpha_k^{(j)}) \in (0, \infty)^k$ are the Dirichlet parameters for class $j$. Combining likelihoods $P(\hat{\mathbf{p}}(X) \mid Y)$ with priors $P(Y)$ expressing the overall class distribution $\pi \in \Delta_k$, we can use Bayes' rule to express the canonical calibration function $P(Y \mid \hat{\mathbf{p}}(X))$ as follows:

$$\text{generative parametrisation:} \qquad \hat{\mu}_{DirGen}(\mathbf{q}; \alpha, \pi) = (\pi_1 f_1(\mathbf{q}), \ldots, \pi_k f_k(\mathbf{q})) / z \tag{6}$$

where $z = \sum_{j=1}^{k} \pi_j f_j(\mathbf{q})$ is the normaliser, and $f_j$ is the probability density function of the Dirichlet distribution with parameters $\alpha^{(j)}$, gathered into a matrix $\alpha$. It will also be convenient to have two alternative parametrisations of the same family: a linear parametrisation for fitting purposes and a canonical parametrisation for interpretation purposes. These parametrisations are defined as follows:

$$\text{linear parametrisation:} \qquad \hat{\mu}_{DirLin}(\mathbf{q}; \mathbf{W}, \mathbf{b}) = \sigma(\mathbf{W}\ln\mathbf{q} + \mathbf{b}) \tag{7}$$

where $\mathbf{W} \in \mathbb{R}^{k \times k}$ is a $k \times k$ parameter matrix, $\ln$ is a vector function that calculates the natural logarithm component-wise and $\mathbf{b} \in \mathbb{R}^k$ is a parameter vector of length $k$;

$$\text{canonical parametrisation:} \qquad \hat{\mu}_{Dir}(\mathbf{q}; \mathbf{A}, \mathbf{c}) = \sigma\left(\mathbf{A}\ln\frac{\mathbf{q}}{1/k} + \ln\mathbf{c}\right) \tag{8}$$

where each column in the k-by-k matrix $\mathbf{A} \in [0, \infty)^{k \times k}$ with non-negative entries contains at least one value 0, division of $\mathbf{q}$ by $1/k$ is component-wise, and $\mathbf{c} \in \Delta_k$ is a probability vector of length $k$.

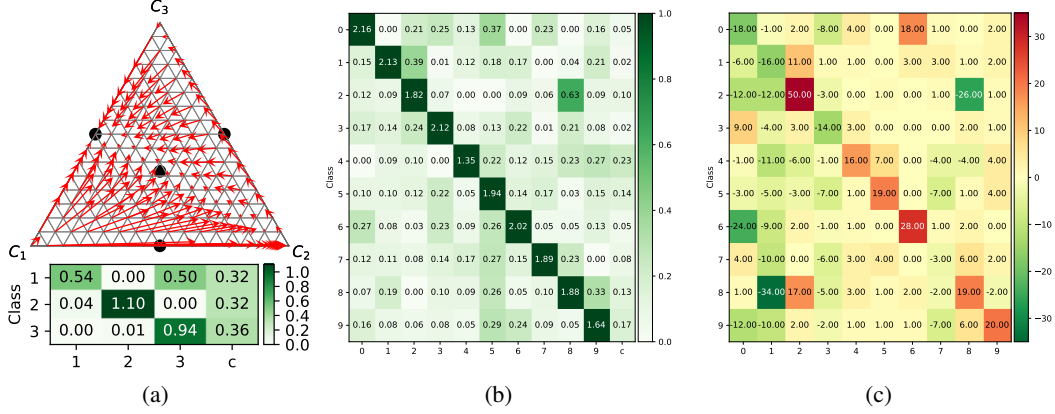

| | (a) | | (b) | | (c) |

Figure 2: Interpretation of Dirichlet calibration maps: (a) calibration map for MLP on the abalone dataset, 4 interpretation points shown by black dots, and canonical parametrisation as a matrix with $\mathbf{A}, \mathbf{c}$; (b) canonical parametrisation of a map on SVHN_convnet; (c) changes to the confusion matrix after applying this calibration map.

**Theorem 1** (Equivalence of generative, linear and canonical parametrisations)**.** *The parametric families $\hat{\mu}_{DirGen}(\mathbf{q}; \alpha, \pi)$, $\hat{\mu}_{DirLin}(\mathbf{q}; \mathbf{W}, \mathbf{b})$ and $\hat{\mu}_{Dir}(\mathbf{q}; \mathbf{A}, \mathbf{c})$ are equal, i.e. they contain exactly the same calibration maps.*

*Proof.* All proofs are given in the Supplemental Material. □

The benefit of the linear parametrisation is that it can be easily implemented as (additional) layers in a neural network: a logarithmic transformation followed by a fully connected layer with softmax activation. Out of the three parametrisations only the canonical parametrisation is unique, in the sense that any function in the Dirichlet calibration map family can be represented by a single pair of matrix $\mathbf{A}$ and vector $\mathbf{c}$ satisfying the requirements set by the canonical parametrisation $\hat{\mu}_{Dir}(\mathbf{q}; \mathbf{A}, \mathbf{c})$.

**Interpretability.** In addition to providing uniqueness, the canonical parametrisation is to some extent interpretable. As demonstrated in the proof of Thm. 1 provided in the Supplemental Material, the linear parametrisation $\mathbf{W}, \mathbf{b}$ obtained after fitting can be easily transformed into the canonical parametrisation by $a_{ij} = w_{ij} - \min_i w_{ij}$ and $\mathbf{c} = \sigma(\mathbf{W} \ln \mathbf{u} + \mathbf{b})$, where $\mathbf{u} = (1/k, \ldots, 1/k)$. In the canonical parametrisation, increasing the value of element $a_{ij}$ in matrix $\mathbf{A}$ increases the calibrated probability of class $i$ (and decreases the probabilities of all other classes), with effect size depending on the uncalibrated probability of class $j$. E.g., element $a_{3,9} = 0.63$ of Fig.2b increases class 2 probability whenever class 8 has high predicted probability, modifying decision boundaries and resulting in 26 less confusions of class 2 for 8 as seen in Fig.2c. Looking at the matrix $\mathbf{A}$ and vector $\mathbf{c}$, it is hard to know the effect of the calibration map without performing the computations. However, at $k + 1$ 'interpretation points' this is (approximately) possible. One of these is the centre of the probability simplex, which maps to $\mathbf{c}$. The other $k$ points are vectors where one value is (almost) zero and the other values are equal, summing up to 1. Figure 2a shows the 3+1 interpretation points in an example for $k = 3$, where each arrow visualises the result of calibration (end of arrow) at a particular point (beginning of arrow). The result of calibration map at the interpretation points in the centres of sides (facets) is each determined by a single column of $\mathbf{A}$ only. The $k$ columns of matrix $\mathbf{A}$ and the vector $\mathbf{c}$ determine, respectively, the behaviour of the calibration map near the $k + 1$ points

$$\left( \varepsilon, \frac{1-\varepsilon}{k-1}, \ldots, \frac{1-\varepsilon}{k-1} \right), \ldots, \left( \frac{1-\varepsilon}{k-1}, \ldots, \frac{1-\varepsilon}{k-1}, \varepsilon \right), \text{ and } \left( \frac{1}{k}, \ldots, \frac{1}{k} \right)$$

The first $k$ points are infinitesimally close to the centres of facets of the probability simplex, and the last point is the centre of the whole simplex. For 3 classes these 4 points have been visualised on the simplex in Fig. 2a. The Dirichlet calibration map $\hat{\mu}_{Dir}(\mathbf{q}; \mathbf{A}, \mathbf{c})$ transforms these $k + 1$ points into:

$$\left( \varepsilon^{a_{11}}, \ldots, \varepsilon^{a_{k1}} \right) / z_1, \ldots, \left( \varepsilon^{a_{1k}}, \ldots, \varepsilon^{a_{kk}} \right) / z_k, \text{ and } (c_1, \ldots, c_k)$$

where $z_i$ are normalising constants, and $a_{ij}, c_j$ are elements of the matrix $\mathbf{A}$ and vector $\mathbf{c}$, respectively. However, the effect of each parameter goes beyond the interpretation points and also changes classification decision boundaries. This can be seen for the calibration map for a model `SVHN_convnet` in Fig. 2b where larger off-diagonal coefficients $a_{ij}$ often result in a bigger change in the confusion matrix as seen in Fig. 2c (particularly in the 3rd row and 9th column).

**Relationship to other families.** For 2 classes, the Dirichlet calibration map family coincides with the beta calibration map family [15]. Although temperature scaling has been defined on logits $\mathbf{z}$, it can be expressed in terms of the model outputs $\hat{\mathbf{p}} = \sigma(\mathbf{z})$ as well. It turns out that temperature scaling maps all belong to the Dirichlet family, with $\hat{\mu}_{TempS}(\mathbf{q};t) = \hat{\mu}_{DirLin}(\mathbf{q}; \frac{1}{t}\mathbf{I}, \mathbf{0})$, where $\mathbf{I}$ is the identity matrix and $\mathbf{0}$ is the zero vector (see Prop.1 in the Supplemental Material). The Dirichlet calibration family is also related to the matrix scaling family $\hat{\mu}_{MatS}(\mathbf{z};\mathbf{W},\mathbf{b}) = \sigma(\mathbf{Wz}+\mathbf{b})$ proposed by [9] alongside with temperature scaling. Both families use a fully connected layer with softmax activation, but the crucial difference is in the inputs to this layer. Matrix scaling uses logits $\mathbf{z}$, while the linear parametrisation of Dirichlet calibration uses log-transformed probabilities $\ln(\hat{\mathbf{p}}) = \ln(\sigma(\mathbf{z}))$. As softmax followed by log-transform is losing information, matrix scaling has an informational advantage over Dirichlet calibration on deep neural networks, which we will turn back to in the experiments section.

**Fitting and ODIR regularisation.** The results of [9] showed poor performance for matrix scaling (with ECE, log-loss, error rate), leading the authors to the conclusion that "[a]ny calibration model with tens of thousands (or more) parameters will overfit to a small validation set, even when applying regularization". We agree that some overfitting happens, but in our experiments a simple L2 regularisation suffices on many non-neural models, whereas for other cases including deep neural nets we propose a novel ODIR (Off-Diagonal and Intercept Regularisation) scheme, which is efficient enough in fighting overfitting to make both Dirichlet calibration and matrix scaling outperform temperature scaling on many occasions, including cases with 100 classes and hence 10100 parameters. Fitting of Dirichlet calibration maps is performed by minimising log-loss, and by adding ODIR regularisation terms to the loss function as follows:

$$ L = \frac{1}{n}\sum_{i=1}^{n} logloss\Big(\hat{\mu}_{DirLin}(\hat{\mathbf{p}}(\mathbf{x}_i);\mathbf{W},\mathbf{b}),y_i\Big) + \lambda \cdot \left(\frac{1}{k(k-1)}\sum_{i \neq j} w_{ij}^2\right) + \mu \cdot \left(\frac{1}{k}\sum_{j} b_j^2\right) $$

where $(\mathbf{x}_i, y_i)$ are validation instances and $w_{ij}, b_j$ are elements of $\mathbf{W}$ and $\mathbf{b}$, respectively, and $\lambda, \mu$ are hyper-parameters tunable with internal cross-validation on the validation data. The intuition is that the diagonal is allowed to freely follow the biases of classes, whereas the intercept is regularised separately from the off-diagonal elements due to having different scales (additive vs. multiplicative).

**Implementation details.** Implementation of Dirichlet calibration is straightforward in standard deep neural network frameworks (we used Keras [5] in the neural experiments). Alternatively, it is also possible to use the Newton–Raphson method on the L2 regularised objective function, which is constructed by applying multinomial logistic regression with $k$ features (log-transformed predicted class probabilities). Both the gradient and Hessian matrix can be calculated either analytically or using automatic differentiation libraries (e.g. JAX [2]). Such implementations normally yield faster convergence given the convexity of the multinomial logistic loss, which is a better choice with a small number of target classes (tractable Hessian). One can also simply adopt existing implementations of logistic regression (e.g. scikit-learn) with the log transformed predicted probabilities. If the uncalibrated model outputs zero probability for some class, then this needs to be clipped to a small positive number (we used $2.2e^{-308}$, the smallest positive usable number for the type `float64` in Python).

## 4 Experiments

The main goals of our experiments are to: (1) compare performance of Dirichlet calibration with other general-purpose calibration methods on a wide range of datasets and classifiers; (2) compare Dirichlet calibration with temperature scaling on several deep neural networks and study the effectiveness of ODIR regularisation; and (3) study whether the neural-specific calibration methods outperform general-purpose calibration methods due to the information loss going from logits to softmax outputs.

Table 1: Calibration methods ranked for **p-cw-ECE** (Friedman's test significant with p-value $1.19e^{-118}$).

|  | Uncal | DirL2 | DirODIR | Beta | TempS | VecS | Isot | FreqB | WidthB |
|---|---|---|---|---|---|---|---|---|---|
| adas | 7.4 | 3.1 | **3.0** | 5.1 | 6.8 | 3.7 | 4.1 | 6.0 | 5.7 |
| forest | 6.1 | 4.9 | 4.7 | **3.2** | 4.4 | 4.3 | 3.6 | 8.0 | 5.8 |
| knn | 7.7 | **2.0** | 4.2 | 5.0 | 6.1 | 4.6 | 3.0 | 7.0 | 5.5 |
| lda | 7.3 | **2.9** | 4.1 | 3.5 | 6.0 | 3.8 | 3.7 | 7.9 | 5.8 |
| logistic | 6.9 | **3.0** | 4.3 | 3.6 | 5.1 | 4.3 | 4.1 | 8.0 | 5.6 |
| mlp | 5.0 | **2.5** | 5.2 | 2.9 | 4.3 | 4.5 | 5.0 | 8.6 | 6.9 |
| nbayes | 8.2 | **1.6** | 2.3 | 4.9 | 7.0 | 3.2 | 4.5 | 6.8 | 6.5 |
| qda | 7.5 | **2.2** | 3.6 | 3.8 | 6.0 | 4.6 | 3.7 | 8.2 | 5.5 |
| svc-linear | 6.2 | **3.0** | 4.4 | 4.0 | 4.5 | 4.0 | 4.8 | 8.5 | 5.5 |
| svc-rbf | 5.5 | 3.8 | 4.1 | 4.4 | 5.0 | 4.8 | **3.4** | 8.3 | 5.8 |
| tree | 3.7 | 4.6 | **3.6** | 5.0 | 4.2 | 4.4 | 4.2 | 8.0 | 7.3 |
| avg rank | 6.50 | **3.07** | 3.97 | 4.12 | 5.40 | 4.21 | 4.01 | 7.75 | 5.98 |

Table 2: Rankings for **log-loss** (Friedman's test significant with p-value $1.58e^{-103}$).

|  | Uncal | DirL2 | DirODIR | Beta | TempS | VecS | Isot | FreqB | WidthB |
|---|---|---|---|---|---|---|---|---|---|
| adas | 8.4 | 2.1 | **2.0** | 4.7 | 7.8 | 5.1 | 4.9 | 4.8 | 5.1 |
| forest | 7.0 | 5.7 | 4.4 | **3.1** | 3.9 | 3.4 | 5.8 | 6.8 | 4.8 |
| knn | 8.2 | 2.8 | 4.9 | 6.4 | 7.2 | 5.4 | **2.7** | 4.4 | 3.0 |
| lda | 7.4 | **1.9** | 3.0 | 3.5 | 5.6 | 4.0 | 7.2 | 7.1 | 5.2 |
| logistic | 7.6 | **1.7** | 2.5 | 3.2 | 5.3 | 4.0 | 8.1 | 7.6 | 5.0 |
| mlp | 4.5 | **2.7** | 4.5 | 3.1 | 3.5 | 4.0 | 8.0 | 8.6 | 6.2 |
| nbayes | 8.6 | **1.4** | 2.5 | 4.7 | 6.2 | 4.4 | 5.8 | 4.9 | 6.5 |
| qda | 7.5 | **2.2** | 3.0 | 3.8 | 5.4 | 4.5 | 6.7 | 7.3 | 4.6 |
| svc-linear | 6.7 | **1.9** | 2.3 | 3.7 | 4.4 | 3.7 | 8.1 | 8.1 | 6.0 |
| svc-rbf | 7.6 | 4.0 | 3.4 | 3.7 | 5.7 | **2.9** | 6.0 | 5.4 | 6.3 |
| tree | 6.2 | 3.6 | 5.7 | 6.2 | 7.4 | 6.7 | **2.0** | 4.0 | 3.2 |
| avg rank | 7.24 | **2.73** | 3.48 | 4.21 | 5.66 | 4.38 | 5.93 | 6.28 | 5.10 |

## 4.1 Calibration of non-neural models

**Experimental setup.** Calibration methods were compared on 21 UCI datasets (*abalone*, *balance-scale*, *car*, *cleveland*, *dermatology*, *glass*, *iris*, *landsat-satellite*, *libras-movement*, *mfeat-karhunen*, *mfeat-morphological*, *mfeat-zernike*, *optdigits*, *page-blocks*, *pendigits*, *segment*, *shuttle*, *vehicle*, *vowel*, *waveform-5000*, *yeast*) with 11 classifiers: multiclass logistic regression (*logistic*), naive Bayes (*nbayes*), random forest (*forest*), adaboost on trees (*adas*), linear discriminant analysis (*lda*), quadratic discriminant analysis (*qda*), decision tree (*tree*), K-nearest neighbours (*knn*), multilayer perceptron (*mlp*), support vector machine with linear (*svc-linear*) and RBF kernel (*svc-rbf*).

In each of the $21 \times 11 = 231$ settings[1] we performed nested cross-validation to evaluate 8 calibration methods: one-vs-rest isotonic calibration (**Isot**) which learns an isotonic calibration map on each class vs rest separately and renormalises the individual calibration map outputs to add up to one at test time; one-vs-rest equal-width binning (**WidthB**) where one-vs-rest calibration maps predict the empirical proportion of labels in each of the equal-width bins of the range $[0,1]$; one-vs-rest equal-frequency binning (**FreqB**) constructing bins with equal numbers of instances; one-vs-rest beta calibration (**Beta**); temperature scaling (**TempS**); vector scaling (**VectS**), which restricts the matrix scaling family, fixing off-diagonal elements to zero [9] (here applied on log-transformed class probabilities instead of logits); and Dirichlet Calibration with L2 (**DirL2**) and with ODIR (**DirODIR**) regularisation. We used 3-fold internal cross-validation to train the calibration maps within the 5 times 5-fold external cross-validation. Following [24], the 3 calibration maps learned in the internal cross-validation were all used as an ensemble by averaging their predictions. For calibration methods with hyperparameters we used the training folds of the classifier to choose the hyperparameter values with the lowest log-loss.

We used 8 evaluation measures: accuracy, log-loss, Brier score, maximum calibration error (MCE), confidence-ECE (conf-ECE), classwise-ECE (cw-ECE), as well as significance measures p-conf-ECE and p-cw-ECE evaluating how often the respective ECE measures are not significantly higher than when assuming calibration. For p-conf-ECE and p-cw-ECE we used significance level $\alpha = 0.05$ in the test of [25] as explained in Section 2, and counted the proportion of significance tests accepting the model being calibrated out of $5 \times 5$ cases of external cross-validation. With each of the 8 evaluation measures we ranked the methods on each of the $21 \times 11$ tasks and performed Friedman tests to find statistical differences [7]. When the p-value of the Friedman test was under 0.005 we performed a post-hoc one-tailed Bonferroni-Dunn test to obtain Critical Differences (CDs) which indicated the minimum ranking difference to consider the methods significantly different. Further details of the experimental setup are provided in the Supplemental Material.

**Results.** The results showed that Dirichlet with ODIR or L2 regularisation was the best calibration method based on log-loss, and p-cw-ECE, and in the group of best calibrators for the other measures except MCE (WidthB was the best for MCE, with all other calibration methods in the second-best group). After grouping the results by the classifier learning algorithm, the average ranks with respect to log-loss are shown in Table 2, and with respect to p-cw-ECE in Table 1. The critical difference diagram for p-cw-ECE is presented in Fig. 3a. Fig. 3b shows the average p-cw-ECE for each calibration method across all datasets and shows how frequently the statistical test accepted

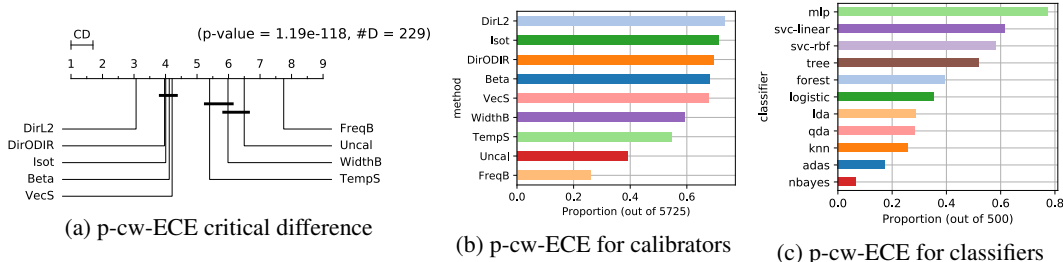

(a) p-cw-ECE critical difference      (b) p-cw-ECE for calibrators      (c) p-cw-ECE for classifiers

Figure 3: Summarised results for **p-cw-ECE**: (a) CD diagram; (b) proportion of times each calibrator was calibrated ($\alpha = 0.05$); (c) proportion of times each classifier was already calibrated ($\alpha = 0.05$).

the null hypothesis of classifier being calibrated (higher p-cw-ECE is better). The results show that DirL2 was considered calibrated on more than 70% of the p-cw-ECE tests. An evaluation of classwise-calibration without post-hoc calibration is given in Fig. 3c. Note that svc-linear and svc-rbf have an unfair advantage because their *sklearn* implementation uses Platt scaling with 3-fold internal cross-validation to provide probabilities.

Supplemental material contains the final ranking tables and CD diagrams for every metric, an analysis of the best calibrator hyperparameters, and a more detailed comparison of the classwise calibration for the 11 classifiers.

## 4.2 Calibration of deep neural networks

**Experimental setup.** We used 3 datasets (CIFAR-10, CIFAR-100 and SVHN), training 11 deep convolutional neural nets with various architectures: ResNet 110 [10], ResNet 110 SD [12], ResNet 152 SD [12], DenseNet 40 [11], WideNet 32 [28], LeNet 5 [18], and acquiring 3 pretrained models from [4]. For the latter we set aside 5,000 test instances for fitting the calibration map. On other models we followed [9], setting aside 5,000 training instances (6,000 in SVHN) for calibration purposes and training the models as in the original papers. For calibration methods with hyperparameters we used 5-fold cross-validation on the validation set to find optimal regularisation parameters. We used all 5 calibration models with the optimal hyperparameter values by averaging their predictions as in [24].

Among general-purpose calibration methods we compared 2 variants of Dirichlet calibration (with L2 regularisation and with ODIR) against temperature scaling (as discussed in Section 3, it can equivalently act on probabilities instead of logits and is therefore general-purpose). Other methods from our non-neural experiment were not included, as these were outperformed by temperature scaling in the experiments of [9]. Among methods that use logits (neural-specific calibration methods) we included matrix scaling with ODIR regularisation, and vector scaling. As reported by [9], the non-regularised matrix scaling performed very poorly and was not included in our comparisons. Full details and source code for training the models are in the Supplemental Material.

**Results.** Tables 3 and 4 show that the best among three general-purpose calibration methods depends heavily on the model and dataset. Both variants of Dirichlet calibration (with L2 and with ODIR) outperformed temperature scaling in most cases on CIFAR-10. On CIFAR-100, Dir-L2 is poor, but Dir-ODIR outperforms TempS in cw-ECE, showing the effectiveness of ODIR regularisation. However, this comes at the expense of minor increase in log-loss. According to the average rank across all deep net experiments, Dir-ODIR is best, but without statistical significance.

The full comparison including calibration methods that use logits confirms that information loss going from logits to softmax outputs has an effect and MS-ODIR (matrix scaling with ODIR) outperforms Dir-ODIR in 8 out of 14 cases on cw-ECE and 11 out of 14 on log-loss. However, the effect is numerically usually very small, as average relative reduction of cw-ECE and log-loss is less than 1% (compared to the average relative reduction of over 30% from the uncalibrated model). According to the average rank on cw-ECE the best method is vector scaling, but this comes at the expense of increased log-loss. According to the average rank on log-loss the best method is MS-ODIR, while its cw-ECE is on average bigger than for vector scaling by 2%.

Table 3: Scores and ranking of calibration methods for **cw-ECE**.

| | Uncal | general-purpose calibrators | | | calibrators using logits | |
|---|---|---|---|---|---|---|
| | | TempS | Dir-L2 | Dir-ODIR | VecS | MS-ODIR |
| c10_convnet | $0.104_6$ | $0.044_4$ | $0.043_2$ | $0.045_5$ | $\mathbf{0.043_1}$ | $0.044_3$ |
| c10_densenet40 | $0.114_6$ | $0.040_5$ | $\mathbf{0.034_1}$ | $0.037_4$ | $0.036_2$ | $0.037_3$ |
| c10_lenet5 | $0.198_6$ | $0.171_5$ | $\mathbf{0.052_1}$ | $0.059_4$ | $0.057_2$ | $0.059_3$ |
| c10_resnet110 | $0.098_6$ | $0.043_5$ | $\mathbf{0.032_1}$ | $0.039_4$ | $0.037_3$ | $0.036_2$ |
| c10_resnet110_SD | $0.086_6$ | $0.031_4$ | $0.031_5$ | $0.029_3$ | $0.027_2$ | $\mathbf{0.027_1}$ |
| c10_resnet_wide32 | $0.095_6$ | $0.048_5$ | $0.032_3$ | $0.029_2$ | $0.032_4$ | $\mathbf{0.029_1}$ |
| c100_convnet | $0.424_6$ | $\mathbf{0.227_1}$ | $0.402_5$ | $0.240_3$ | $0.241_4$ | $0.240_2$ |
| c100_densenet40 | $0.470_6$ | $0.187_2$ | $0.330_5$ | $\mathbf{0.186_1}$ | $0.189_3$ | $0.191_4$ |
| c100_lenet5 | $0.473_6$ | $0.385_5$ | $0.219_4$ | $0.213_2$ | $\mathbf{0.203_1}$ | $0.214_3$ |
| c100_resnet110 | $0.416_6$ | $0.201_3$ | $0.359_5$ | $\mathbf{0.186_1}$ | $0.194_2$ | $0.203_4$ |
| c100_resnet110_SD | $0.375_6$ | $0.203_4$ | $0.373_5$ | $0.189_3$ | $\mathbf{0.170_1}$ | $0.186_2$ |
| c100_resnet_wide32 | $0.420_6$ | $0.184_4$ | $0.333_5$ | $0.180_2$ | $\mathbf{0.171_1}$ | $0.180_3$ |
| SVHN_convnet | $0.159_6$ | $0.038_4$ | $0.043_5$ | $0.026_2$ | $\mathbf{0.025_1}$ | $0.027_3$ |
| SVHN_resnet152_SD | $0.019_2$ | $\mathbf{0.018_1}$ | $0.022_6$ | $0.020_3$ | $0.021_5$ | $0.021_4$ |
| Average rank | 5.71 | 3.71 | 3.79 | 2.79 | 2.29 | 2.71 |

Table 4: Scores and ranking of calibration methods for **log-loss**.

| | Uncal | general-purpose calibrators | | | calibrators using logits | |
|---|---|---|---|---|---|---|
| | | TempS | Dir-L2 | Dir-ODIR | VecS | MS-ODIR |
| c10_convnet | $0.391_6$ | $\mathbf{0.195_1}$ | $0.197_4$ | $0.195_2$ | $0.197_5$ | $0.196_3$ |
| c10_densenet40 | $0.428_6$ | $0.225_5$ | $\mathbf{0.220_1}$ | $0.224_4$ | $0.223_3$ | $0.222_2$ |
| c10_lenet5 | $0.823_6$ | $0.800_5$ | $0.744_2$ | $0.744_3$ | $0.747_4$ | $\mathbf{0.743_1}$ |
| c10_resnet110 | $0.358_6$ | $0.209_5$ | $\mathbf{0.203_1}$ | $0.205_3$ | $0.206_4$ | $0.204_2$ |
| c10_resnet110_SD | $0.303_6$ | $0.178_5$ | $0.177_4$ | $0.176_3$ | $0.175_2$ | $\mathbf{0.175_1}$ |
| c10_resnet_wide32 | $0.382_6$ | $0.191_5$ | $0.185_4$ | $0.182_2$ | $0.183_3$ | $\mathbf{0.182_1}$ |
| c100_convnet | $1.641_6$ | $\mathbf{0.942_1}$ | $1.189_5$ | $0.961_2$ | $0.964_4$ | $0.961_3$ |
| c100_densenet40 | $2.017_6$ | $1.057_2$ | $1.253_5$ | $1.059_4$ | $1.058_3$ | $\mathbf{1.051_1}$ |
| c100_lenet5 | $2.784_6$ | $2.650_5$ | $2.595_4$ | $2.490_2$ | $2.516_3$ | $\mathbf{2.487_1}$ |
| c100_resnet110 | $1.694_6$ | $1.092_3$ | $1.212_5$ | $1.096_4$ | $1.089_2$ | $\mathbf{1.074_1}$ |
| c100_resnet110_SD | $1.353_6$ | $0.942_3$ | $1.198_5$ | $0.945_4$ | $\mathbf{0.923_1}$ | $0.927_2$ |
| c100_resnet_wide32 | $1.802_6$ | $0.945_3$ | $1.087_5$ | $0.953_4$ | $0.937_2$ | $\mathbf{0.933_1}$ |
| SVHN_convnet | $0.205_6$ | $0.151_5$ | $0.142_3$ | $0.138_2$ | $0.144_4$ | $\mathbf{0.138_1}$ |
| SVHN_resnet152_SD | $0.085_6$ | $\mathbf{0.079_1}$ | $0.085_5$ | $0.080_2$ | $0.081_4$ | $0.081_3$ |
| Average rank | 6.0 | 3.5 | 3.79 | 2.93 | 3.14 | 1.64 |

As the difference between MS-ODIR and vector scaling was on some models quite small, we further investigated the importance of off-diagonal coefficients in MS-ODIR. For this we introduced a new model MS-ODIR-zero, obtained from the respective MS-ODIR model by replacing the off-diagonal entries with zeroes. In 6 out of 14 cases (c10_convnet, c10_densenet40, c10_resnet110_SD, c100_convnet, c100_resnet110_SD, SVHN_resnet152_SD) MS-ODIR-zero and MS-ODIR had almost identical performance (difference in log-loss of less than 0.0001), indicating that ODIR regularisation had forced the off-diagonal entries to practically zero. However, MS-ODIR-zero was significantly worse in the remaining 8 out of 14 cases, indicating that the learned off-diagonal coefficients in MS-ODIR were meaningful. In all those cases MS-ODIR outperformed VecS in log-loss. To eliminate the potential explanation that this could be due to random chance, we retrained these networks on 2 more train-test splits (except for the pretrained SVHN_convnet). In all the reruns MS-ODIR remained better than VecS, confirming that it is important to model the pairwise effects between classes in these cases. Detailed results have been presented in the Supplemental Material.

## 5 Conclusion

In this paper we proposed a new parametric general-purpose multiclass calibration method called Dirichlet calibration, which is a natural extension of the two-class beta calibration. Dirichlet calibration is easy to implement as a layer in a neural net, or as multinomial logistic regression on log-transformed class probabilities. Its parameters provide insight into the biases of the model. While derived from Dirichlet-distributed likelihoods, it *does not assume* that the probability vectors are actually Dirichlet-distributed within each class, similarly as logistic calibration (Platt scaling) does not assume that the scores are Gaussian-distributed, while it can be derived from Gaussian likelihoods.

Comparisons with other general-purpose calibration methods across *21 datasets $\times$ 11 models* showed best or tied best performance for Dirichlet calibration on all 8 evaluation measures. Evaluation with our proposed classwise-ECE measure quantifies how calibrated the predicted probabilities are on all classes, not only on the most likely predicted class as with the commonly used (confidence-)ECE. On neural networks we advance the state-of-the-art by introducing the ODIR regularisation scheme for matrix scaling and Dirichlet calibration, leading these to outperform temperature scaling on many deep neural networks.

Interestingly, on many deep nets Dirichlet calibration learns a map which is very close to being in a temperature scaling family. This raises a fundamental theoretical question of which neural architectures and training methods result in a classifier with its canonical calibration function contained in the temperature scaling family. But even in those cases Dirichlet calibration can become useful after any kind of dataset shift, learning an interpretable calibration map to reveal the shift and recalibrate the predictions for the new context.

Deriving calibration maps from Dirichlet distributions opens up the possibility of using other distributions of the exponential family to obtain new calibration maps designed for various score types, as well as investigating scores coming from mixtures of distributions inside each class.

## Acknowledgements

The work of MKu and MKä was supported by the Estonian Research Council under grant PUT1458. The work of MPN and HS was supported by the SPHERE Next Steps Project funded by the UK Engineering and Physical Sciences Research Council (EPSRC), Grant EP/R005273/1. The work of PF and HS was supported by The Alan Turing Institute under EPSRC, Grant EP/N510129/1.

## Footnotes

[1]Naive Bayes and QDA ran too long on dataset *shuttle*, leaving us with a total of 229 sets of results.

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
