[Supplementary Material · NeurIPS2019___Dirichlet_Calibration_supplementary.pdf]

# Beyond temperature scaling:
# Obtaining well-calibrated multi-class probabilities with Dirichlet calibration
## *Supplementary material*

**Meelis Kull**
Department of Computer Science
University of Tartu
meelis.kull@ut.ee

**Miquel Perello-Nieto**
Department of Computer Science
University of Bristol
miquel.perellonieto@bris.ac.uk

**Markus Kängsepp**
Department of Computer Science
University of Tartu
markus.kangsepp@ut.ee

**Telmo Silva Filho**
Department of Statistics
Universidade Federal da Paraíba
telmo@de.ufpb.br

**Hao Song**
Department of Computer Science
University of Bristol
hao.song@bristol.ac.uk

**Peter Flach**
Department of Computer Science
University of Bristol and
The Alan Turing Institute
peter.flach@bristol.ac.uk

## Contents

# A    Source code

The instructions and code for the experiments can be found on https://dirichletcal.github.io/.

# B    Proofs

**Theorem 1** (Equivalence of generative, linear and canonical parametrisations)**.** *The parametric families $\hat{\boldsymbol{\mu}}_{DirGen}(\mathbf{q}; \boldsymbol{\alpha}, \boldsymbol{\pi})$, $\hat{\boldsymbol{\mu}}_{DirLin}(\mathbf{q}; \mathbf{W}, \mathbf{b})$ and $\hat{\boldsymbol{\mu}}_{Dir}(\mathbf{q}; \mathbf{A}, \mathbf{c})$ are equal, i.e. they contain exactly the same calibration maps.*

*Proof.* We will prove that:

1. every function in $\hat{\boldsymbol{\mu}}_{DirGen}(\mathbf{q}; \boldsymbol{\alpha}, \boldsymbol{\pi})$ belongs to $\hat{\boldsymbol{\mu}}_{DirLin}(\mathbf{q}; \mathbf{W}, \mathbf{b})$;

2. every function in $\hat{\boldsymbol{\mu}}_{DirLin}(\mathbf{q}; \mathbf{W}, \mathbf{b})$ belongs to $\hat{\boldsymbol{\mu}}_{Dir}(\mathbf{q}; \mathbf{A}, \mathbf{c})$;

3. every function in $\hat{\boldsymbol{\mu}}_{Dir}(\mathbf{q}; \mathbf{A}, \mathbf{c})$ belongs to $\hat{\boldsymbol{\mu}}_{DirGen}(\mathbf{q}; \boldsymbol{\alpha}, \boldsymbol{\pi})$.

**1.**    Consider a function $\hat{\mu}(\mathbf{q}) = \hat{\boldsymbol{\mu}}_{DirGen}(\mathbf{q}; \boldsymbol{\alpha}, \boldsymbol{\pi})$. Let us start with an observation that any vector $\mathbf{x} = (x_1, \ldots, x_k) \in (0, \infty)^k$ with only positive elements can be renormalised to add up to $1$ using the expression $\boldsymbol{\sigma}(\ln(\mathbf{x}))$, since:

$$\boldsymbol{\sigma}(\ln(\mathbf{x})) = \exp(\ln(\mathbf{x}))/(\sum_i \exp(\ln(x_i))) = \mathbf{x}/(\sum_i x_i)$$

where $\exp$ is an operator applying exponentiation element-wise. Therefore,

$$\hat{\mu}(\mathbf{q}) = \boldsymbol{\sigma}(\ln(\pi_1 f_1(\mathbf{q}), \ldots, \pi_k f_k(\mathbf{q})))$$

where $f_i(\mathbf{q})$ is the probability density function of the distribution $Dir(\boldsymbol{\alpha}^{(i)})$ where $\boldsymbol{\alpha}^{(i)}$ is the $i$-th row of matrix $\boldsymbol{\alpha}$. Hence, $f_i(\mathbf{q}) = \frac{1}{B(\boldsymbol{\alpha}^{(i)})} \prod_{j=1}^{k} q_j^{\alpha_{ij}-1}$, where $B(\cdot)$ denotes the multivariate beta function. Let us define a matrix $\mathbf{W}$ and vector $\mathbf{b}$ as follows:

$$w_{ij} = \alpha_{ij} - 1, \qquad b_i = \ln(\pi_i) - \ln(B(\boldsymbol{\alpha}^{(i)}))$$

with $w_{ij}$ and $\alpha_{ij}$ denoting elements of matrices $\mathbf{W}$ and $\boldsymbol{\alpha}$, respectively, and $b_i, \pi_i$ denoting elements of vectors $\mathbf{b}$ and $\boldsymbol{\pi}$. Now we can write

$$\ln(\pi_i f_i(\mathbf{q})) = \ln(\pi_i) - \ln(B(\boldsymbol{\alpha}^{(i)})) + \ln \prod_{j=1}^{k} q_j^{\alpha_{ij}-1}$$

$$= \ln(\pi_i) - \ln(B(\boldsymbol{\alpha}^{(i)})) + \sum_{j=1}^{k} (\alpha_{ij} - 1) \ln(q_j)$$

$$= b_i + \sum_{j=1}^{k} w_{ij} \ln(q_j)$$

and substituting this back into $\hat{\mu}(\mathbf{q})$ we get:

$$\hat{\mu}(\mathbf{q}) = \boldsymbol{\sigma}(\ln(\pi_1 f_1(\mathbf{q}), \ldots, \pi_k f_k(\mathbf{q})))$$
$$= \boldsymbol{\sigma}(\mathbf{b} + \mathbf{W}\ln(\mathbf{q})) = \hat{\mu}_{DirLin}(\mathbf{q}; \mathbf{W}, \mathbf{b})$$

**2.**    Consider a function $\hat{\mu}(\mathbf{q}) = \hat{\boldsymbol{\mu}}_{DirLin}(\mathbf{q}; \mathbf{W}, \mathbf{b})$. Let us define a matrix $\mathbf{A}$ and vector $\mathbf{c}$ as follows:

$$a_{ij} = w_{ij} - \min_i w_{ij}, \qquad \mathbf{c} = \boldsymbol{\sigma}(\mathbf{W} \ln \mathbf{u} + \mathbf{b})$$

with $a_{ij}$ and $w_{ij}$ denoting elements of matrices $\mathbf{A}$ and $\mathbf{W}$, respectively, and $\mathbf{u} = (1/k, \ldots, 1/k)$ is a column vector of length $k$. Note that $\mathbf{A}\mathbf{x} = \mathbf{W}\mathbf{x} + const_1$ and $\ln \boldsymbol{\sigma}(\mathbf{x}) = \mathbf{x} + const_2$ for any $x$ where $const_1$ and $const_2$ are constant vectors (all elements are equal), but the constant depends on

x. Taking into account that $\boldsymbol{\sigma}(\mathbf{v} + const) = \boldsymbol{\sigma}(\mathbf{v})$ for any vector $\mathbf{v}$ and constant vector $const$, we obtain:

$$\hat{\mu}_{Dir}(\mathbf{q}; \mathbf{A}, \mathbf{c}) = \boldsymbol{\sigma}(\mathbf{A} \ln \frac{\mathbf{q}}{1/k} + \ln \mathbf{c}) = \boldsymbol{\sigma}(\mathbf{W} \ln \frac{\mathbf{q}}{1/k} + const_1 + \ln \mathbf{c})$$

$$= \boldsymbol{\sigma}(\mathbf{W} \ln \mathbf{q} - \mathbf{W} \ln \mathbf{u} + const_1 + \ln \boldsymbol{\sigma}(\mathbf{W} \ln \mathbf{u} + \mathbf{b}))$$

$$= \boldsymbol{\sigma}(\mathbf{W} \ln \mathbf{q} - \mathbf{W} \ln \mathbf{u} + const_1 + \mathbf{W} \ln \mathbf{u} + \mathbf{b} + const_2)$$

$$= \boldsymbol{\sigma}(\mathbf{W} \ln \mathbf{q} + \mathbf{b} + const_1 + const_2) = \boldsymbol{\sigma}(\mathbf{W} \ln \mathbf{q} + \mathbf{b}) = \hat{\mu}_{DirLin}(\mathbf{q}; \mathbf{W}, \mathbf{b})$$

$$= \hat{\mu}(\mathbf{q})$$

**3.** Consider a function $\hat{\mu}(\mathbf{q}) = \hat{\mu}_{Dir}(\mathbf{q}; \mathbf{A}, \mathbf{c})$. Let us define a matrix $\boldsymbol{\alpha}$, vector $\mathbf{b}$ and vector $\boldsymbol{\pi}$ as follows:

$$\alpha_{ij} = a_{ij} + 1, \qquad \mathbf{b} = \ln \mathbf{c} - \mathbf{A} \ln \mathbf{u}, \qquad \pi_i = \exp(b_i) \cdot B(\boldsymbol{\alpha}^{(i)})$$

with $\alpha_{ij}$ and $a_{ij}$ denoting elements of matrices $\boldsymbol{\alpha}$ and $\mathbf{A}$, respectively, and $\mathbf{u} = (1/k, \dots, 1/k)$ is a column vector of length $k$. We can now write:

$$\hat{\mu}(\mathbf{q}) = \hat{\mu}_{Dir}(\mathbf{q}; \mathbf{A}, \mathbf{c}) = \boldsymbol{\sigma}(\mathbf{A} \ln \frac{\mathbf{q}}{1/k} + \ln \mathbf{c}) = \boldsymbol{\sigma}(\mathbf{A} \ln \mathbf{q} - \mathbf{A} \ln \mathbf{u} + \ln \mathbf{c})$$

$$= \boldsymbol{\sigma}((\boldsymbol{\alpha} - 1)\ln \mathbf{q} + \mathbf{b})$$

Element $i$ in the vector within the softmax is equal to:

$$\sum_{j=1}^{k} (\boldsymbol{\alpha}_{ij} - 1) \ln(q_j) + b_j = \sum_{j=1}^{k} (\boldsymbol{\alpha}_{ij} - 1) \ln(q_j) + \ln(\pi_i \cdot \frac{1}{B(\boldsymbol{\alpha}^{(i)})})$$

$$= \ln(\pi_i \cdot \frac{1}{B(\boldsymbol{\alpha}^{(i)})} \prod_{j=1}^{k} q_j^{\boldsymbol{\alpha}_{ij}-1})$$

$$= \ln(\pi_i \cdot f_i(\boldsymbol{\alpha}^{(i)}))$$

and therefore:

$$\hat{\mu}(\mathbf{q}) = \boldsymbol{\sigma}((\boldsymbol{\alpha} - 1)\ln(\mathbf{q}) + \mathbf{b}) = \boldsymbol{\sigma}(\ln(\pi_i \cdot f_i(\boldsymbol{\alpha}^{(i)}))) = \hat{\boldsymbol{\mu}}_{DirGen}(\mathbf{q}; \boldsymbol{\alpha}, \boldsymbol{\pi})$$

$\square$

The following proposition proves that temperature scaling can be viewed as a general-purpose calibration method, being a special case within the Dirichlet calibration map family.

**Proposition 1.** *Let us denote the temperature scaling family by $\hat{\mu}'_{TempS}(\mathbf{z}; t) = \boldsymbol{\sigma}(\mathbf{z}/t)$ where $\mathbf{z}$ are the logits. Then for any $t$, temperature scaling can be expressed as*

$$\hat{\mu}'_{TempS}(\mathbf{z}; t) = \hat{\mu}_{DirLin}(\boldsymbol{\sigma}(\mathbf{z}); \frac{1}{t}\mathbf{I}, \mathbf{0})$$

*where $\mathbf{I}$ is the identity matrix and $\mathbf{0}$ is the vector of zeros.*

*Proof.* Let us first observe that for any $\mathbf{x} \in \mathbb{R}^k$ there exists a constant vector $const$ (all elements are equal) such that $\ln \boldsymbol{\sigma}(\mathbf{x}) = \mathbf{x} + const$. Furthermore, $\boldsymbol{\sigma}(\mathbf{v} + const) = \boldsymbol{\sigma}(\mathbf{v})$ for any vector $\mathbf{v}$ and any constant vector $const$. Therefore,

$$\hat{\mu}_{DirLin}(\boldsymbol{\sigma}(\mathbf{z}); \frac{1}{t}\mathbf{I}, \mathbf{0}) = \boldsymbol{\sigma}(\frac{1}{t}\mathbf{I} \ln \boldsymbol{\sigma}(\mathbf{z})))$$

$$= \boldsymbol{\sigma}(\frac{1}{t}\mathbf{I}(\mathbf{z} + const))$$

$$= \boldsymbol{\sigma}(\frac{1}{t}\mathbf{I}\mathbf{z} + \frac{1}{t}\mathbf{I}\,const)$$

$$= \boldsymbol{\sigma}(\mathbf{z}/t + const')$$

$$= \boldsymbol{\sigma}(\mathbf{z}/t)$$

$$= \hat{\mu}'_{TempS}(\mathbf{z}; t)$$

where $const' = \frac{1}{t}\mathbf{I}\,const$ is a constant vector as a product of a diagonal matrix with a constant vector. $\square$

# C  Dirichlet calibration

In this section we show some examples of reliability diagrams and other plots that can help to understand the representational power of Dirichlet calibration compared with other calibration methods.

## C.1  Reliability diagram examples

We will look at two examples of reliability diagrams on the original classifier and after applying 6 calibration methods. Figure 1 shows the first example for the 3 class classification dataset *balance-scale* and the classifier MLP. This figure shows the confidence-reliability diagram in the first column and the classwise-reliability diagrams in the other columns. Figure 1a shows how posterior probabilities from the MLP have small gaps between the true class proportions and the predicted means. This visualisation may indicate that the original classifier is already well calibrated. However, when we separate the reliability diagram per class, we notice that the predictions for the first class are underconfident, as indicated by high proportions of the true class at low mean predictions. On the other hand, the predictions on classes 2 and 3 are overconfident in the regions of posterior probabilities compressed between $[0.2, 0.5]$ while being underconfident in higher regions.

Table 1: Averaged results for the confidence-ECE and classwise-ECE metrics of 6 calibrators applied on an MLP trained on the *balance-scale* dataset.

|  | DirL2 | Beta | FreqB | Isot | WidB | TempS | Uncal |
|---|---|---|---|---|---|---|---|
| conf-ECE | $\mathbf{0.04}_1$ | $0.05_3$ | $0.13_7$ | $0.05_2$ | $0.08_6$ | $0.05_4$ | $0.08_5$ |
| cw-ECE | $0.12_2$ | $0.13_3$ | $0.29_7$ | $\mathbf{0.12}_1$ | $0.17_5$ | $0.15_4$ | $0.20_6$ |

The following subfigures show how the different calibration methods try to reduce ECE, occasionally increasing the error. As can be seen in Table 1, Dirichlet L2 and One-vs.Rest isotonic regression obtain the lowest ECE while One-vs.Rest frequency binning makes the original calibration worse. Looking at Figure 1i it is possible to see how temperature scaling manages to reduce the overall overconfidence in the higher range of probabilities for classes 2 and 3, but makes the situation worse in the interval $[0.2, 0.6]$. However, it manages to reduce the overall ECE.

In the second example we show 3 calibration methods for a 4 class classification problem (car dataset) applied on the scores of an Adaboost SAMME classifier. Figure 2 shows one reliability diagram per class ($C_1$ *acceptable*, $C_2$ *good*, $C_3$ *unacceptable*, and $C_4$ *very good*).

From this Figure we can see that the uncalibrated model is underconfident for classes 1, 2 and 3, showing posterior probabilities never higher than $0.7$, while having true class proportions higher than $0.7$ in the mentioned interval. We can see that after applying some of the calibration models the posterior probabilities reach higher probability values. As can be seen in Table 2, Dirichlet L2

Table 2: Averaged results for the confidence-ECE and classwise-ECE metrics of 6 calibrators applied on an Adaboost SAMME trained on the *car* dataset.

|  | DirL2 | Beta | FreqB | Isot | WidB | TempS | Uncal |
|---|---|---|---|---|---|---|---|
| conf-ECE | $\mathbf{0.07}_1$ | $0.10_4$ | $0.12_5$ | $0.07_2$ | $0.09_3$ | $0.14_7$ | $0.14_6$ |
| cw-ECE | $0.18_2$ | $0.23_3$ | $0.29_5$ | $\mathbf{0.18}_1$ | $0.25_4$ | $0.32_7$ | $0.29_6$ |

and One-vs.Rest Isotonic Regression obtain the lowest ECE while Temperature Scaling makes the original calibration worse. Figure 2d shows how Dirichlet calibration with L2 regularisation achieved the largest spread of probabilities, also reducing the error mean gap with the predictions and the true class proportions. On the other hand, temperature scaling reduced ECE for class 1, but hurt the overall performance for the other classes.

A more detailed depiction of the previous reliability diagrams can be seen in Figure 3. In this case, the posterior probabilities are not introduced in bins, but a boxplot summarises their full distribution. The first observation here is, for the *good* and *very good* classes, the uncalibrated model tends to predict probability vectors with small variance, i.e. the outputs do not change much among different instances. Among the calibration approaches, temperature scaling still maintains this low level of

(a) Uncalibrated

(b) Uncalibrated per class

(c) OvR Frequency Binning

(d) One-vs.-Rest Frequency Binning per class

(e) OvR Width Binning

(f) One-vs.-Rest Width Binning calibration per class

(g) OvR Isotonic calibration

(h) One-vs.-Rest Isotonic calibration per class

(i) Temperature Scaling

(j) Temperature Scaling

(k) OvR Beta calibration

(l) One-vs.-Rest Beta calibration per class

(m) Dirichlet L2

(n) Dirichlet with L2 regularisation per class

Figure 1: Confidence-reliability diagrams in the first column and classwise-reliability diagrams in the remaining columns, for a real experiment with the multilayer perceptron classifier on the balance-scale dataset and a subset of the calibrators. All the test partitions from the 5 times 5-fold-cross-validation have been aggregated to draw every plot.

Figure 2: Reliability diagrams per class for a real experiment with the classifier Ada boost SAMME on the car dataset and 3 calibrators. The test partitions from the 5 times 5-fold-cross-validation have been aggregated to draw every plot.

variance, while both isotonic and Dirichlet L2 manage to show a higher variance on the outputs. While this observation cannot be justified here without quantitative analysis, another observation clearly shows an advantage of using Dirichlet L2. For the *acceptable* class, only Dirichlet L2 is capable of providing the highest mean probability for the correct class, while the other three methods tend to put higher probability mass on the *unacceptable* class on average.

# D   Experimental setup

In this section we provide the detailed description of the experimental setup on a variety of non-neural classifiers and datasets. While our implementation of Dirichlet calibration is based on standard Newton-Raphson with multinomial logistic loss and L2 regularisation, as mentioned at the end of Section 3, existing implementations of logistic regression (e.g. scikit-learn) with the log transformed predicted probabilities can also be easily applied.

## D.1   Datasets and performance estimation

The full list of datasets, and a brief description of each one including the number of samples, features and classes is presented in Table 3.

Figure 4 shows how every dataset was divided in order to get an estimated performance for every combination of dataset, classifier and calibrator. Each dataset was divided using 5 times 5-fold-cross-validation to create 25 test partitions. For each of the 25 partitions the corresponding training set was divided further with a 3-fold-cross-validation for wich the bigger portions were used to train the classifiers (and validate the calibratiors if they had hyperparameters), and the small portion was used to train the calibrators. The 3 calibrators trained in the inner 3-folds were used to predict the corresponding test partition, and their predictions were averaged in order to obtain better estimates of their performance with the 7 different metrics (accuracy, Brier score, log-loss, maximum calibration

|                     |                    |                          |                 |
|:-------------------:|:------------------:|:------------------------:|:---------------:|
| (a) Uncalibrated    | (b) Isotonic       | (c) Temperature scaling  | (d) Dirichlet L2 |

Figure 3: Effect of Dirichlet Calibration on the scores of Ada boost SAMME on the *car* dataset which is composed of $4$ classes (*acceptable*, *good*, *unacceptable*, and *very good*). The whiskers of each box indicate the 5th and 95th percentile, the notch around the median indicates the confidence interval. The green error bar to the right of each box indicates one standard deviation on each side of the mean. In each subfigure, the first boxplot corresponds to the posterior probabilities for the samples of class 1, divided in 4 boxes representing the posterior probabilities for each class. A good classifier should have the highest posterior probabilities in the box corresponding to the true class. In Figure 3a it is possible to see that the first class (*acceptable*) is missclassified as belonging to the third class (*unacceptable*) with high probability values, while Dirichlet Calibration is able to alleviate that problem. Also, for the second and fourth true classes (*good*, and *very good*) the original classifier uses a reduced domain of probabilities (indicative of underconfidence), while Dirichlet calibration is able to spread these probabilities with more meaningful values (as indicated by a reduction of the calibration losses; See Figure 2).

error, confidence-ECE, classwise-ECE and the p test statistic of the ECE metrics). Finally, the 25 resulting measures were averaged.

## D.2 Full example of statistical analysis

The following is a full example of how the final rankings and statistical tests are computed. For this example, we will focus on the metric log-loss, and we will start with the naive Bayes classifier. Table 4 shows the estimated log-loss by averaging the 5-times 5-fold cross-validation log-losses of the inner 3-fold aggregated predictions. The sub-indices are the ranking of every calibrator for each dataset (ties in the ranking share the averaged rank). The resulting table of sub-indices is used to compute the Friedman test statistic, resulting in a value of 97.9 and a p-value of $1.14e^{-17}$ indicating statistical difference between the different calibration methods. The last row contains the average ranks of the full table, which is shown in the corresponding critical difference diagram in Figure 5a. The critical difference uses the Bonferroni-Dunn one-tailed statistical test to compute the minimum ranking distance that is shown in the Figure, indicating that for this particular classifier and metric the Dirichlet calibrator with L2 regularisation is significantly better than the other methods, with the exception of Dirichlet with ODIR regularisation.

The same process is applied to each of the 11 classifiers for every metric. Table 6 shows the final average results of all classifiers. Notice that the row corresponding to naive Bayes has the rounded average rankings from Figure 5a.

| dataset | n_samples | n_features | n_classes |
|---|---|---|---|
| abalone | 4177 | 8 | 3 |
| balance-scale | 625 | 4 | 3 |
| car | 1728 | 6 | 4 |
| cleveland | 297 | 13 | 5 |
| dermatology | 358 | 34 | 6 |
| glass | 214 | 9 | 6 |
| iris | 150 | 4 | 3 |
| landsat-satellite | 6435 | 36 | 6 |
| libras-movement | 360 | 90 | 15 |
| mfeat-karhunen | 2000 | 64 | 10 |
| mfeat-morphological | 2000 | 6 | 10 |
| mfeat-zernike | 2000 | 47 | 10 |
| optdigits | 5620 | 64 | 10 |
| page-blocks | 5473 | 10 | 5 |
| pendigits | 10992 | 16 | 10 |
| segment | 2310 | 19 | 7 |
| shuttle | 101500 | 9 | 7 |
| vehicle | 846 | 18 | 4 |
| vowel | 990 | 10 | 11 |
| waveform-5000 | 5000 | 40 | 3 |
| yeast | 1484 | 8 | 10 |

Table 3: Datasets used for the large-scale empirical comparison.

Figure 4: Partitions of each dataset in order to estimate out-of-sample measures.

| | Uncal | DirL2 | DirODIR | Beta | TempS | VecS | Isot | FreqB | WidthB |
|---|---|---|---|---|---|---|---|---|---|
| abalone | $1.95_9$ | $\mathbf{0.89_1}$ | $0.89_2$ | $0.89_5$ | $0.89_6$ | $0.89_4$ | $0.90_7$ | $0.89_3$ | $0.92_8$ |
| balance-sc | $0.48_9$ | $0.22_2$ | $\mathbf{0.22_1}$ | $0.31_4$ | $0.41_8$ | $0.23_3$ | $0.38_7$ | $0.36_6$ | $0.35_5$ |
| car | $1.57_9$ | $0.38_2$ | $\mathbf{0.38_1}$ | $0.59_6$ | $0.92_8$ | $0.43_3$ | $0.56_5$ | $0.55_4$ | $0.67_7$ |
| cleveland | $2.72_9$ | $1.02_2$ | $\mathbf{1.00_1}$ | $1.30_6$ | $1.36_8$ | $1.08_3$ | $1.32_7$ | $1.12_4$ | $1.14_5$ |
| dermatolog | $2.57_9$ | $\mathbf{0.20_1}$ | $0.41_7$ | $0.36_5$ | $0.31_3$ | $0.44_8$ | $0.33_4$ | $0.23_2$ | $0.40_6$ |
| glass | $2.93_9$ | $\mathbf{1.12_1}$ | $1.57_6$ | $1.62_8$ | $1.35_4$ | $1.48_5$ | $1.58_7$ | $1.13_3$ | $1.12_2$ |
| iris | $0.13_4$ | $\mathbf{0.11_1}$ | $0.12_3$ | $0.13_5$ | $0.12_2$ | $0.13_6$ | $0.30_8$ | $0.33_9$ | $0.21_7$ |
| landsat-sa | $3.87_9$ | $\mathbf{0.35_1}$ | $0.36_2$ | $0.55_4$ | $0.61_7$ | $0.55_3$ | $0.58_5$ | $0.58_6$ | $0.74_8$ |
| libras-mov | $4.99_9$ | $\mathbf{0.94_1}$ | $1.73_7$ | $1.32_3$ | $1.16_2$ | $1.33_4$ | $2.04_8$ | $1.66_6$ | $1.43_5$ |
| mfeat-karh | $0.45_9$ | $\mathbf{0.20_1}$ | $0.20_2$ | $0.23_3$ | $0.24_5$ | $0.23_4$ | $0.39_8$ | $0.38_7$ | $0.29_6$ |
| mfeat-morp | $1.78_9$ | $\mathbf{0.71_1}$ | $0.79_2$ | $0.91_7$ | $1.01_8$ | $0.86_5$ | $0.85_4$ | $0.82_3$ | $0.88_6$ |
| mfeat-zern | $1.73_9$ | $0.60_2$ | $\mathbf{0.59_1}$ | $0.71_3$ | $0.76_5$ | $0.71_4$ | $0.85_8$ | $0.82_6$ | $0.84_7$ |
| optdigits | $3.23_9$ | $0.46_3$ | $0.46_2$ | $0.54_6$ | $0.83_7$ | $0.52_5$ | $\mathbf{0.43_1}$ | $0.47_4$ | $0.84_8$ |
| page-block | $0.76_9$ | $\mathbf{0.17_1}$ | $0.17_3$ | $0.19_4$ | $0.48_8$ | $0.21_6$ | $0.17_2$ | $0.20_5$ | $0.21_7$ |
| pendigits | $1.29_9$ | $0.19_2$ | $\mathbf{0.19_1}$ | $0.46_3$ | $0.53_7$ | $0.46_4$ | $0.47_5$ | $0.48_6$ | $0.58_8$ |
| segment | $1.39_9$ | $\mathbf{0.28_1}$ | $0.31_2$ | $0.46_4$ | $0.53_7$ | $0.46_5$ | $0.47_6$ | $0.46_3$ | $0.56_8$ |
| vehicle | $2.34_9$ | $\mathbf{0.98_1}$ | $0.99_2$ | $1.08_5$ | $1.16_7$ | $1.07_4$ | $1.17_8$ | $1.04_3$ | $1.10_6$ |
| vowel | $0.83_5$ | $\mathbf{0.55_1}$ | $0.62_2$ | $0.80_3$ | $0.85_6$ | $0.80_4$ | $1.04_8$ | $1.06_9$ | $0.89_7$ |
| waveform-5 | $0.80_9$ | $\mathbf{0.33_1}$ | $0.33_2$ | $0.37_4$ | $0.43_7$ | $0.35_3$ | $0.39_6$ | $0.38_5$ | $0.46_8$ |
| yeast | $5.12_9$ | $1.26_2$ | $\mathbf{1.22_1}$ | $1.33_6$ | $2.06_8$ | $1.32_5$ | $1.29_3$ | $1.31_4$ | $1.43_7$ |
| avg rank | 8.55 | **1.40** | 2.50 | 4.70 | 6.15 | 4.40 | 5.85 | 4.90 | 6.55 |

Table 4: Ranking of calibration methods applied on the classifier nbayes with the measure=loss(Friedman statistic test = 9.79E+01, p-value = 1.14E-17)

(a) Average over all datasets for Naive Bayes classifier

(b) Average over all classifiers

Figure 5: Critical Difference diagrams for the averaged ranking results of the metric Log-loss.

# E  Results

In this Section we present all the final results, including ranking tables for every metric, critical difference diagrams, the best hyperparameters selected for Dirichlet calibration with L2 regularisation, Frequency binning and Width binning; a comparison of how calibrated the 11 classifiers are, and additional results on deep neural networks.

## E.1  Final ranking tables for all metrics

We present here all the final ranking tables for all metrics (Tables 5, 6, 7, 8, 9, 10, 11, and 12). For each ranking, a lower value is indicative of a better metric value (eg. a higher accuracy corresponds to a lower ranking, while a lower log-loss corresponds to a lower ranking as well). Additional details on how to interpret the tables can be found in Section D.2.

Table 5: Rankings for Accuracy

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

6). The results showed Dirichlet L2 as the best calibration method for the measures accuracy, log-loss and p-cw-ece with statistical significance (See Figures 6a 6c, and 6h), and in the group of the best calibration methods in the rest of the metrics with statistical significance, but no difference within the group. It is worth mentioning that Figure 6c showed statistical difference between Dirichlet L2, OvR Beta, OvR width binning, and the rest of the calibrators in one group; in the mentioned order.

### E.3 Best calibrator hyperparameters

Figure 8 shows the best hyperparameters for every inner 3-fold-cross-validation. Dirichlet L2 (Figure 8a) shows a preference for regularisation hyperparameter $\lambda = 1e^{-3}$ and lower values. Our current minimum regularisation value of $1e^{-7}$ is also being selected multiple times, indicating that lower values may be optimal in several occasions. However, this fact did not seem to hurt the overall good results in our experiments. One-vs.-Rest frequency binning tends to prefer 10 bins of equal number of samples, while One-vs.Rest width binning prefers 5 equal sized bins (See Figures 8b and 8c respectively).

### E.4 Comparison of classifiers

In this Section we compare all the classifiers without post-hoc calibration on 17 of the datasets; from the total of 21 datasets *shuttle*, *yeast*, *mfeat-karhunen* and *libras-movement* were removed from this analysis as at least one classifier was not able to complete the experiment.

Figure 9 shows the Critical Difference diagram for all the 8 metrics. In particular, the MLP and the SVC with linear kernel are always in the group with the higher rankings and never in the lowest. Similarly, random forest is consistently in the best group, but in the worst group as well in 4 of the measures. SVC with radial basis kernel is in the best group 6 times, but 3 times in the worst. On the other hand, naive Bayes and Adaboost SAMME are consistently in the worst group and never in the best one. The rest of the classifiers did not show a clear ranking position.

Figures 10b and 10a show the proportion of times each classifier passed the p-conf-ECE and p-cw-ECE statistical test for all datasets and cross-validation folds.

### E.5 Deep neural networks

In this section, we provide further discussion about results from the deep networks experiments. These are given in the form of critical difference diagrams (Figure 11) and tables (Tables 13-20) both including the following measures: error rate, log-loss, Brier score, maximum calibration error (MCE), confidence-ECE (conf-ECE), classwise-ECE (cw-ECE), as well as significance measures p-conf-ECE and p-cw-ECE.

In addition, Table 21 compares MS-ODIR and vector scaling on log-loss. On the table, we also added MS-ODIR-zero which was obtained from the respective MS-ODIR model by replacing the off-diagonal entries with zeroes. Each experiment is replicated three times with different splits on datasets. This is done to compare the stability of the methods. In each replication, the best scoring model is written in bold.

Finally, Figure 12 shows that temperature scaling systematically under-estimates class 4 probabilities on the model `c10_resnet_wide32` on CIFAR-10.

Figure 6: Critical difference of the average of multiclass classifiers.

Figure 7: Proportion of times each calibrator passes a calibration p-test with a p-value higher than 0.05.

(a) Dirichlet L2 $\lambda$     (b) OvR Frequency binning #bins     (c) OvR Width binning #bins

Figure 8: Histogram of the selected hyperparameters during the inner 3-fold-cross-validation

(a) Accuracy

(b) Log-loss

(c) Brier score

(d) Maximum Calibration Error

(e) conf-ece

(f) cw-ece

(g) p-conf-ece

(h) p-cw-ece

Figure 9: Critical difference of uncalibrated classifiers.

(a) p-conf-ece      (b) p-cw-ece

Figure 10: Proportion of times each classifier is already calibrated with different p-tests.

(a) Error Rate      (b) Brier score

(c) Log-loss score      (d) Maximum calibration error

(e) conf-ece      (f) cw-ece

(g) p-conf-ece      (h) p-cw-ece

Figure 11: Critical difference of the deep neural networks.

Table 13: Scores and ranking of calibration methods for **log-loss**.

| | Uncal | general-purpose calibrators | | | calibrators using logits | |
| --- | --- | --- | --- | --- | --- | --- |
| | | TempS | Dir-L2 | Dir-ODIR | VecS | MS-ODIR |
| c10_convnet | $0.39098_6$ | $\mathbf{0.19497_1}$ | $0.19692_4$ | $0.19536_2$ | $0.19743_5$ | $0.19634_3$ |
| c10_densenet40 | $0.42821_6$ | $0.22509_5$ | $\mathbf{0.22048_1}$ | $0.22371_4$ | $0.22270_3$ | $0.22240_2$ |
| c10_lenet5 | $0.82326_6$ | $0.80031_5$ | $0.74418_2$ | $0.74441_3$ | $0.74704_4$ | $\mathbf{0.74262_1}$ |
| c10_resnet110 | $0.35827_6$ | $0.20926_5$ | $\mathbf{0.20303_1}$ | $0.20511_3$ | $0.20595_4$ | $0.20375_2$ |
| c10_resnet110_SD | $0.30325_6$ | $0.17760_5$ | $0.17694_4$ | $0.17608_3$ | $0.17549_2$ | $\mathbf{0.17537_1}$ |
| c10_resnet_wide32 | $0.38170_6$ | $0.19148_5$ | $0.18464_4$ | $0.18203_2$ | $0.18276_3$ | $\mathbf{0.18165_1}$ |
| c100_convnet | $1.64120_6$ | $\mathbf{0.94162_1}$ | $1.18945_5$ | $0.96121_2$ | $0.96369_4$ | $0.96141_3$ |
| c100_densenet40 | $2.01740_6$ | $1.05713_2$ | $1.25293_5$ | $1.05909_4$ | $1.05831_3$ | $\mathbf{1.05084_1}$ |
| c100_lenet5 | $2.78365_6$ | $2.64979_5$ | $2.59482_4$ | $2.48951_2$ | $2.51590_3$ | $\mathbf{2.48670_1}$ |
| c100_resnet110 | $1.69371_6$ | $1.09169_3$ | $1.21239_5$ | $1.09607_4$ | $1.08916_2$ | $\mathbf{1.07370_1}$ |
| c100_resnet110_SD | $1.35250_6$ | $0.94214_3$ | $1.19837_5$ | $0.94477_4$ | $\mathbf{0.92341_1}$ | $0.92731_2$ |
| c100_resnet_wide32 | $1.80215_6$ | $0.94453_3$ | $1.08711_5$ | $0.95288_4$ | $0.93650_2$ | $\mathbf{0.93273_1}$ |
| SVHN_convnet | $0.20460_6$ | $0.15142_5$ | $0.14246_3$ | $0.13791_2$ | $0.14388_4$ | $\mathbf{0.13760_1}$ |
| SVHN_resnet152_SD | $0.08542_6$ | $\mathbf{0.07861_1}$ | $0.08463_5$ | $0.08038_2$ | $0.08124_4$ | $0.08100_3$ |
| avg rank | 6.0 | 3.5 | 3.79 | 2.93 | 3.14 | 1.64 |

Figure 12: Reliability diagrams of c10_resnet_wide32 on CIFAR-10: (a) classwise-reliability for class 4 after temperature scaling; (b) classwise-reliability for class 4 after Dirichlet calibration.

Table 14: Scores and ranking of calibration methods for **Brier score**.

|  | Uncal | general-purpose calibrators | | | calibrators using logits | |
|---|---|---|---|---|---|---|
|  |  | TempS | Dir-L2 | Dir-ODIR | VecS | MS-ODIR |
| c10_convnet | $0.01090_6$ | $\textbf{0.00952}_1$ | $0.00969_5$ | $0.00955_3$ | $0.00958_4$ | $0.00953_2$ |
| c10_densenet40 | $0.01274_6$ | $0.01100_4$ | $0.01102_5$ | $0.01097_2$ | $0.01097_3$ | $\textbf{0.01097}_1$ |
| c10_lenet5 | $0.03788_6$ | $0.03748_5$ | $0.03510_2$ | $0.03511_3$ | $0.03523_4$ | $\textbf{0.03502}_1$ |
| c10_resnet110 | $0.01102_6$ | $0.00979_4$ | $0.00979_5$ | $0.00977_2$ | $0.00978_3$ | $\textbf{0.00976}_1$ |
| c10_resnet110_SD | $0.00981_6$ | $0.00874_4$ | $0.00877_5$ | $0.00867_3$ | $0.00867_2$ | $\textbf{0.00866}_1$ |
| c10_resnet_wide32 | $0.01047_6$ | $0.00924_5$ | $0.00909_4$ | $\textbf{0.00888}_1$ | $0.00891_3$ | $0.00889_2$ |
| c100_convnet | $0.00425_5$ | $\textbf{0.00358}_1$ | $0.00441_6$ | $0.00358_2$ | $0.00362_4$ | $0.00361_3$ |
| c100_densenet40 | $0.00491_6$ | $0.00401_3$ | $0.00468_5$ | $0.00400_2$ | $0.00403_4$ | $\textbf{0.00400}_1$ |
| c100_lenet5 | $0.00813_6$ | $0.00792_5$ | $0.00786_4$ | $0.00760_2$ | $0.00767_3$ | $\textbf{0.00760}_1$ |
| c100_resnet110 | $0.00453_6$ | $0.00392_3$ | $0.00438_5$ | $0.00391_2$ | $0.00393_4$ | $\textbf{0.00391}_1$ |
| c100_resnet110_SD | $0.00418_5$ | $0.00367_4$ | $0.00456_6$ | $0.00364_3$ | $\textbf{0.00360}_1$ | $0.00361_2$ |
| c100_resnet_wide32 | $0.00432_6$ | $0.00355_4$ | $0.00401_5$ | $0.00354_3$ | $0.00352_2$ | $\textbf{0.00351}_1$ |
| SVHN_convnet | $0.00776_6$ | $0.00598_5$ | $0.00555_3$ | $\textbf{0.00530}_1$ | $0.00561_4$ | $0.00532_2$ |
| SVHN_resnet152_SD | $0.00297_3$ | $\textbf{0.00291}_1$ | $0.00305_6$ | $0.00293_2$ | $0.00299_5$ | $0.00298_4$ |
| avg rank | 5.64 | 3.5 | 4.71 | 2.21 | 3.29 | 1.64 |

Table 15: Scores and ranking of calibration methods for **confidence-ECE**.

|  | Uncal | general-purpose calibrators | | | calibrators using logits | |
|---|---|---|---|---|---|---|
|  |  | TempS | Dir-L2 | Dir-ODIR | VecS | MS-ODIR |
| c10_convnet | $0.04760_6$ | $0.01065_5$ | $0.00769_2$ | $0.00960_4$ | $\textbf{0.00740}_1$ | $0.00782_3$ |
| c10_densenet40 | $0.05500_6$ | $0.00946_2$ | $\textbf{0.00568}_1$ | $0.01097_5$ | $0.01018_4$ | $0.00988_3$ |
| c10_lenet5 | $0.05180_6$ | $0.01665_5$ | $0.01383_3$ | $0.01367_2$ | $\textbf{0.01310}_1$ | $0.01468_4$ |
| c10_resnet110 | $0.04750_6$ | $0.01132_5$ | $\textbf{0.00680}_1$ | $0.01086_3$ | $0.01130_4$ | $0.01059_2$ |
| c10_resnet110_SD | $0.04113_6$ | $\textbf{0.00555}_1$ | $0.00646_4$ | $0.00815_5$ | $0.00579_3$ | $0.00566_2$ |
| c10_resnet_wide32 | $0.04505_6$ | $0.00784_4$ | $\textbf{0.00524}_1$ | $0.00837_5$ | $0.00769_3$ | $0.00727_2$ |
| c100_convnet | $0.17614_6$ | $\textbf{0.01367}_1$ | $0.14347_5$ | $0.02069_3$ | $0.01965_2$ | $0.02660_4$ |
| c100_densenet40 | $0.21156_6$ | $\textbf{0.00902}_1$ | $0.12380_5$ | $0.01138_2$ | $0.01224_3$ | $0.02197_4$ |
| c100_lenet5 | $0.12125_6$ | $0.01499_4$ | $0.01369_2$ | $0.02003_5$ | $\textbf{0.01294}_1$ | $0.01407_3$ |
| c100_resnet110 | $0.18480_6$ | $\textbf{0.02380}_1$ | $0.14535_5$ | $0.02822_4$ | $0.02693_2$ | $0.02735_3$ |
| c100_resnet110_SD | $0.15861_5$ | $\textbf{0.01214}_1$ | $0.15920_6$ | $0.02283_4$ | $0.01296_2$ | $0.02246_3$ |
| c100_resnet_wide32 | $0.18784_6$ | $\textbf{0.01472}_1$ | $0.13509_5$ | $0.01891_3$ | $0.01718_2$ | $0.02581_4$ |
| SVHN_convnet | $0.07755_6$ | $0.01179_4$ | $0.01910_5$ | $0.00997_2$ | $\textbf{0.00934}_1$ | $0.01037_3$ |
| SVHN_resnet152_SD | $0.00862_6$ | $0.00607_4$ | $0.00691_5$ | $\textbf{0.00582}_1$ | $0.00595_2$ | $0.00604_3$ |
| avg rank | 5.93 | 2.79 | 3.57 | 3.43 | 2.21 | 3.07 |

Table 16: Scores and ranking of calibration methods for **classwise-ECE**.

| | Uncal | general-purpose calibrators | | | calibrators using logits | |
|---|---|---|---|---|---|---|
| | | TempS | Dir-L2 | Dir-ODIR | VecS | MS-ODIR |
| c10_convnet | $0.10375_6$ | $0.04423_4$ | $0.04262_2$ | $0.04507_5$ | $\mathbf{0.04259_1}$ | $0.03532_3$ |
| c10_densenet40 | $0.11430_6$ | $0.03977_5$ | $\mathbf{0.03412_1}$ | $0.03687_4$ | $0.03609_2$ | $0.03678_3$ |
| c10_lenet5 | $0.19849_6$ | $0.17141_5$ | $\mathbf{0.05185_1}$ | $0.05891_4$ | $0.05705_2$ | $0.05862_3$ |
| c10_resnet110 | $0.09846_6$ | $0.04344_5$ | $\mathbf{0.03206_1}$ | $0.03950_4$ | $0.03653_3$ | $0.03615_2$ |
| c10_resnet110_SD | $0.08647_6$ | $0.03071_4$ | $0.03148_5$ | $0.02937_3$ | $0.02713_2$ | $\mathbf{0.02681_1}$ |
| c10_resnet_wide32 | $0.09530_6$ | $0.04775_5$ | $0.03153_3$ | $0.02947_2$ | $0.03164_4$ | $\mathbf{0.02921_1}$ |
| c100_convnet | $0.42414_6$ | $\mathbf{0.22683_1}$ | $0.40185_5$ | $0.24041_3$ | $0.24063_4$ | $0.23958_2$ |
| c100_densenet40 | $0.47026_6$ | $0.18664_2$ | $0.32985_5$ | $\mathbf{0.18630_1}$ | $0.18879_3$ | $0.19112_4$ |
| c100_lenet5 | $0.47264_6$ | $0.38481_5$ | $0.21865_4$ | $0.21348_2$ | $\mathbf{0.20293_1}$ | $0.21379_3$ |
| c100_resnet110 | $0.41644_6$ | $0.20095_3$ | $0.35885_5$ | $\mathbf{0.18639_1}$ | $0.19442_2$ | $0.20270_4$ |
| c100_resnet110_SD | $0.37518_6$ | $0.20310_4$ | $0.37346_5$ | $0.18895_3$ | $\mathbf{0.17015_1}$ | $0.18552_2$ |
| c100_resnet_wide32 | $0.42027_6$ | $0.18573_4$ | $0.33258_5$ | $0.17951_2$ | $\mathbf{0.17082_1}$ | $0.17966_3$ |
| SVHN_convnet | $0.15935_6$ | $0.03830_4$ | $0.04276_5$ | $0.02638_2$ | $\mathbf{0.02480_1}$ | $0.02750_3$ |
| SVHN_resnet152_SD | $0.01940_2$ | $\mathbf{0.01849_1}$ | $0.02184_6$ | $0.01988_3$ | $0.02120_5$ | $0.02088_4$ |
| avg rank | 5.71 | 3.71 | 3.79 | 2.79 | 2.29 | 2.71 |

Table 17: Scores and ranking of calibration methods for **MCE**.

| | Uncal | general-purpose calibrators | | | calibrators using logits | |
|---|---|---|---|---|---|---|
| | | TempS | Dir-L2 | Dir-ODIR | VecS | MS-ODIR |
| c10_convnet | $0.59173_6$ | $0.23150_4$ | $0.12432_2$ | $0.24830_5$ | $0.12831_3$ | $\mathbf{0.07621_1}$ |
| c10_densenet40 | $0.33396_6$ | $0.09929_2$ | $0.11679_4$ | $\mathbf{0.07858_1}$ | $0.12046_5$ | $0.11297_3$ |
| c10_lenet5 | $0.11281_6$ | $0.09158_3$ | $\mathbf{0.05112_1}$ | $0.09009_2$ | $0.09996_4$ | $0.10061_5$ |
| c10_resnet110 | $0.29580_6$ | $0.23639_4$ | $0.24405_5$ | $\mathbf{0.08331_1}$ | $0.13130_2$ | $0.22678_3$ |
| c10_resnet110_SD | $0.32484_6$ | $\mathbf{0.07823_1}$ | $0.23064_5$ | $0.13309_3$ | $0.14276_4$ | $0.08422_2$ |
| c10_resnet_wide32 | $0.37215_4$ | $\mathbf{0.07060_1}$ | $0.49283_6$ | $0.41567_5$ | $0.26539_3$ | $0.26372_2$ |
| c100_convnet | $0.36391_6$ | $0.13689_4$ | $0.23333_5$ | $0.07235_2$ | $\mathbf{0.07043_1}$ | $0.08171_3$ |
| c100_densenet40 | $0.45400_6$ | $\mathbf{0.02213_1}$ | $0.19748_5$ | $0.04074_2$ | $0.04293_3$ | $0.05004_4$ |
| c100_lenet5 | $0.20097_6$ | $0.05836_3$ | $\mathbf{0.05678_1}$ | $0.06774_4$ | $0.05749_2$ | $0.08939_5$ |
| c100_resnet110 | $0.39882_6$ | $0.07099_2$ | $0.20732_5$ | $0.08026_4$ | $0.07354_3$ | $\mathbf{0.06678_1}$ |
| c100_resnet110_SD | $0.48291_6$ | $0.04099_2$ | $0.24578_5$ | $0.05979_3$ | $\mathbf{0.04038_1}$ | $0.06612_4$ |
| c100_resnet_wide32 | $0.45639_6$ | $\mathbf{0.03606_1}$ | $0.19370_5$ | $0.05521_2$ | $0.06605_4$ | $0.06468_3$ |
| SVHN_convnet | $0.30011_5$ | $0.40691_6$ | $\mathbf{0.16154_1}$ | $0.18458_3$ | $0.16312_2$ | $0.18588_4$ |
| SVHN_resnet152_SD | $0.25032_5$ | $\mathbf{0.18244_1}$ | $0.23895_4$ | $0.19649_2$ | $0.23092_3$ | $0.80082_6$ |
| avg rank | 5.71 | 2.5 | 3.86 | 2.79 | 2.86 | 3.29 |

Table 18: Scores and ranking of calibration methods for **error rate (%)**.

| | Uncal | general-purpose calibrators | | | calibrators using logits | |
|---|---|---|---|---|---|---|
| | | TempS | Dir-L2 | Dir-ODIR | VecS | MS-ODIR |
| c10_convnet | $6.18000_2$ | $6.18000_2$ | $6.38000_6$ | $\mathbf{6.12000_1}$ | $6.36000_5$ | $6.32000_4$ |
| c10_densenet40 | $7.58000_5$ | $7.58000_5$ | $\mathbf{7.49000_1}$ | $7.53000_4$ | $7.52000_3$ | $7.50000_2$ |
| c10_lenet5 | $27.26000_5$ | $27.26000_5$ | $\mathbf{25.25000_1}$ | $25.44000_2$ | $25.49000_3$ | $25.50000_4$ |
| c10_resnet110 | $6.44000_1$ | $6.44000_1$ | $6.54000_6$ | $6.49000_4$ | $6.47000_3$ | $6.49000_4$ |
| c10_resnet110_SD | $5.96000_5$ | $5.96000_5$ | $5.90000_4$ | $\mathbf{5.77000_1}$ | $5.83000_3$ | $5.81000_2$ |
| c10_resnet_wide32 | $6.07000_5$ | $6.07000_5$ | $5.94000_4$ | $5.76000_2$ | $\mathbf{5.74000_1}$ | $5.81000_3$ |
| c100_convnet | $26.12000_1$ | $26.12000_1$ | $30.96000_6$ | $26.22000_3$ | $26.56000_4$ | $26.60000_5$ |
| c100_densenet40 | $30.00000_3$ | $30.00000_3$ | $33.48000_6$ | $29.87000_2$ | $30.16000_5$ | $\mathbf{29.61000_1}$ |
| c100_lenet5 | $66.41000_5$ | $66.41000_5$ | $65.97000_4$ | $62.53000_2$ | $63.59000_3$ | $\mathbf{62.44000_1}$ |
| c100_resnet110 | $28.52000_4$ | $28.52000_4$ | $30.04000_6$ | $\mathbf{28.36000_1}$ | $28.40000_2$ | $28.45000_3$ |
| c100_resnet110_SD | $27.17000_4$ | $27.17000_4$ | $31.43000_6$ | $26.96000_3$ | $26.50000_2$ | $\mathbf{26.42000_1}$ |
| c100_resnet_wide32 | $26.18000_4$ | $26.18000_4$ | $27.69000_6$ | $26.07000_3$ | $26.08000_3$ | $\mathbf{26.06000_1}$ |
| SVHN_convnet | $3.82750_5$ | $3.82750_5$ | $3.42811_3$ | $\mathbf{3.34728_1}$ | $3.51845_4$ | $3.37105_2$ |
| SVHN_resnet152_SD | $1.84773_2$ | $1.84773_2$ | $1.90535_6$ | $\mathbf{1.80547_1}$ | $1.87462_4$ | $1.87462_4$ |
| avg rank | 4.14 | 4.14 | 4.64 | 2.11 | 3.25 | 2.71 |

Table 19: Scores and ranking of calibration methods for **p-confidence-ECE**.

| | Uncal | general-purpose calibrators | | | calibrators using logits | |
|---|---|---|---|---|---|---|
| | | TempS | Dir-L2 | Dir-ODIR | VecS | MS-ODIR |
| c10_convnet | $0.0_6$ | $0.032_4$ | $0.363_2$ | $0.019_5$ | $\mathbf{0.461_1}$ | $0.052_3$ |
| c10_densenet40 | $0.0_4$ | $0.002_2$ | $\mathbf{0.525_1}$ | $0.000_4$ | $0.000_4$ | $0.000_4$ |
| c10_lenet5 | $0.0_6$ | $0.008_5$ | $0.027_4$ | $0.084_3$ | $\mathbf{0.155_1}$ | $0.144_2$ |
| c10_resnet110 | $0.0_4$ | $0.000_4$ | $\mathbf{0.246_1}$ | $0.000_4$ | $0.000_4$ | $0.000_4$ |
| c10_resnet110_SD | $0.0_6$ | $0.105_4$ | $\mathbf{0.179_1}$ | $0.003_5$ | $0.114_3$ | $0.124_2$ |
| c10_resnet_wide32 | $0.0_6$ | $0.017_3$ | $\mathbf{0.281_1}$ | $0.005_4$ | $0.005_4$ | $0.076_2$ |
| c100_convnet | $0.0_5$ | $\mathbf{0.174_1}$ | $0.000_5$ | $0.049_2$ | $0.021_3$ | $0.000_5$ |
| c100_densenet40 | $0.0_5$ | $\mathbf{0.817_1}$ | $0.000_5$ | $0.617_2$ | $0.238_3$ | $0.000_5$ |
| c100_lenet5 | $0.0_6$ | $0.153_4$ | $0.217_3$ | $0.001_5$ | $0.395_2$ | $\mathbf{0.422_1}$ |
| c100_resnet110 | $0.0_3$ | $0.000_3$ | $0.000_3$ | $0.000_3$ | $0.000_3$ | $0.000_3$ |
| c100_resnet110_SD | $0.0_4$ | $0.009_2$ | $0.000_4$ | $0.000_4$ | $\mathbf{0.060_1}$ | $0.000_4$ |
| c100_resnet_wide32 | $0.0_4$ | $\mathbf{0.022_1}$ | $0.000_4$ | $0.000_4$ | $0.001_2$ | $0.000_4$ |
| mnist_mlp | $0.0_6$ | $0.616_3$ | $\mathbf{0.948_1}$ | $0.486_4$ | $0.455_5$ | $0.677_2$ |
| SVHN_convnet | $0.0_3$ | $0.000_3$ | $0.000_3$ | $0.000_3$ | $0.000_3$ | $0.000_3$ |
| SVHN_resnet152_SD | $0.0_3$ | $0.000_3$ | $0.000_3$ | $0.000_3$ | $0.000_3$ | $0.000_3$ |
| avg rank | 4.93 | 2.97 | 2.9 | 3.9 | 2.97 | 3.33 |

Table 20: Scores and ranking of calibration methods for **p-classwise-ECE**.

| | Uncal | general-purpose calibrators | | | calibrators using logits | |
|---|---|---|---|---|---|---|
| | | TempS | Dir-L2 | Dir-ODIR | VecS | MS-ODIR |
| c10_convnet | $0.0_6$ | $0.0104_4$ | $\mathbf{0.1276_1}$ | $0.0038_5$ | $0.0340_2$ | $0.0114_3$ |
| c10_densenet40 | $0.0_4$ | $0.0000_4$ | $\mathbf{0.0093_1}$ | $0.0000_4$ | $0.0000_4$ | $0.0000_4$ |
| c10_lenet5 | $0.0_5$ | $0.0000_5$ | $\mathbf{0.6014_1}$ | $0.0390_4$ | $0.1230_2$ | $0.0501_3$ |
| c10_resnet110 | $0.0_4$ | $0.0000_4$ | $\mathbf{0.0088_1}$ | $0.0000_4$ | $0.0000_4$ | $0.0000_4$ |
| c10_resnet110_SD | $0.0_6$ | $0.0058_5$ | $0.0105_3$ | $0.0077_4$ | $0.1816_2$ | $\mathbf{0.2196_1}$ |
| c10_resnet_wide32 | $0.0_5$ | $0.0000_5$ | $0.0096_3$ | $0.0158_2$ | $0.0006_4$ | $\mathbf{0.0249_1}$ |
| c100_convnet | $0.0_4$ | $\mathbf{0.0770_1}$ | $0.0000_4$ | $0.0000_4$ | $0.0000_4$ | $0.0000_4$ |
| c100_densenet40 | $0.0_3$ | $0.0000_3$ | $0.0000_3$ | $0.0000_3$ | $0.0000_3$ | $0.0000_3$ |
| c100_lenet5 | $0.0_3$ | $0.0000_3$ | $0.0000_3$ | $0.0000_3$ | $0.0000_3$ | $0.0000_3$ |
| c100_resnet110 | $0.0_3$ | $0.0000_3$ | $0.0000_3$ | $0.0000_3$ | $0.0000_3$ | $0.0000_3$ |
| c100_resnet110_SD | $0.0_3$ | $0.0000_3$ | $0.0000_3$ | $0.0000_3$ | $0.0000_3$ | $0.0000_3$ |
| c100_resnet_wide32 | $0.0_3$ | $0.0000_3$ | $0.0000_3$ | $0.0000_3$ | $0.0000_3$ | $0.0000_3$ |
| mnist_mlp | $0.0_6$ | $\mathbf{0.5669_1}$ | $0.0842_3$ | $0.0022_5$ | $0.0280_4$ | $0.1178_2$ |
| SVHN_convnet | $0.0_3$ | $0.0000_3$ | $0.0000_3$ | $0.0000_3$ | $0.0000_3$ | $0.0000_3$ |
| SVHN_resnet152_SD | $0.0_3$ | $0.0000_3$ | $0.0000_3$ | $0.0000_3$ | $0.0000_3$ | $0.0000_3$ |
| avg rank | 4.37 | 3.63 | 2.77 | 3.77 | 3.37 | 3.1 |

Table 21: Comparison of MS-ODIR and vector scaling for **log-loss**.

| | Replication 1 | | | Replication 2 | | | Replication 3 | | |
|---|---|---|---|---|---|---|---|---|---|
| | VecS | MS-ODIR | MS-ODIR-zero | VecS | MS-ODIR | MS-ODIR-zero | VecS | MS-ODIR | MS-ODIR-zero |
| c10_convnet | 0.19774 | **0.19632** | 0.19632 | — | — | — | — | — | — |
| c10_densenet40 | **0.22240** | 0.22240 | 0.22240 | 0.21316 | **0.21186** | 0.21366 | 0.21350 | **0.21325** | 0.21327 |
| c10_lenet5 | 0.74688 | **0.74262** | 0.74830 | 0.69392 | **0.69287** | 0.69335 | **0.67955** | 0.67974 | 0.68127 |
| c10_resnet110 | 0.20624 | **0.20375** | 0.20537 | 0.20064 | **0.19803** | 0.20040 | 0.19655 | **0.19536** | 0.19739 |
| c10_resnet110_SD | 0.17545 | **0.17537** | 0.17539 | 0.18123 | **0.18094** | 0.18097 | **0.17799** | 0.17829 | 0.17829 |
| c10_resnet_wide32 | 0.18274 | **0.18165** | 0.18302 | 0.18522 | **0.18364** | 0.18546 | 0.17431 | **0.17274** | 0.17448 |
| c100_convnet | 0.96311 | **0.96141** | 0.96149 | — | — | — | — | — | — |
| c100_densenet40 | 1.05714 | **1.05084** | 1.06804 | 1.06366 | **1.05456** | 1.07107 | 1.07704 | **1.06918** | 1.08559 |
| c100_lenet5 | 2.51695 | **2.48670** | 2.57932 | 2.21546 | **2.20054** | 2.22360 | 2.28054 | **2.27887** | 2.29485 |
| c100_resnet110 | 1.08824 | **1.07370** | 1.10137 | 1.09066 | **1.08267** | 1.11116 | 1.11977 | **1.10672** | 1.13900 |
| c100_resnet110_SD | **0.92275** | 0.92731 | 0.92730 | 0.87758 | **0.87698** | 0.87701 | **0.88523** | 0.88731 | 0.88727 |
| c100_resnet_wide32 | 0.93724 | **0.93273** | 0.94060 | 0.93291 | **0.92531** | 0.94854 | 0.93183 | **0.92439** | 0.94568 |
| SVHN_convnet | 0.14392 | **0.13760** | 0.14507 | — | — | — | — | — | — |
| SVHN_resnet152_SD | 0.08131 | **0.08100** | 0.08100 | 0.12728 | **0.12723** | 0.12639 | 0.12559 | **0.12453** | 0.12381 |