[Reviews · NeurIPS 2019]

Reviewer 1



Originality: the paper is original in many aspects: definition of classwise-calibrated vs. confidence calibrated, as well defining classwise-reliability diagrams, the use of a specific parametric family for calibration which had not been done in previous research, as well as defining ways of visualizing/interpret the results. They also show that using L2 regularization for non-neural models and ODIR regularization for neural models improves the performance of existing methods. All the related work is adequately cited and connections are made between existing work and the proposed method. Clarity: the paper is very well written and easy to follow, enough details are given so that the results can be reproduced. I only think that figure 2 deserves a bit more explanation as what exactly the arrows means and what they mean by "fact centres of the probability simplex". Quality: as far as I could check, the paper is technically sound, with a clear theoretical motivation for the method and empirical evaluation that supports the claim. they also point out limitations of the method and show that for neural models the neural-specific calibrators work better with a certain regularization method. Significance: calibration methods are important for many applications and the results presented in this paper paint a clear picture of the current state-of-the-art both for non-neural and for neural models specifically when classwise-calibration is required.

Reviewer 2



Originality: As far as I know, calibration by modeling parameters in the Dirichlet distribution is novel. The authors also provide very useful analysis of its properties (e.g. equivalence of different parameterizations) and in relation to other calibration families. Quality: The work is solid and convincing. The authors evaluate their calibration algorithm in many different settings in comparison to other algorithms. One aspect that could be improved upon is the evaluation of matrix scaling. Using ODIR, the authors show that matrix scaling can be as effective as vector scaling (and better than temperature scaling) in many settings. However, with sufficiently high regularization strength, off-diagonal values in the matrix could be reduced to 0 and the method becomes identical to vector scaling. Indeed, in Tables 3 and 4, their performance appear to be very similar in all settings. This would contradict the claim the authors make about matrix scaling being effective despite its large number of parameters. Clarity: The paper is well-written and clearly presented, and is in general a smooth read. One minor issue: the average rank in Tables 3 and 4 appear identical. Is this correct? Significance: The topic of multi-class calibration is certainly significant and the authors make an important step towards defining and solving this problem. I can foresee several follow-up papers that expand and improve upon this work.

Reviewer 3



This paper has no major flaw. The main reason I don't vouch for this paper's acceptance is because the method seems limited in both practical usefulness and enlightenment to the reader. It is yet another family of calibration maps, slightly more involved than the simple alternatives of, say, matrix and vector scaling. From reading the exposition and the empirical evaluations, if I wanted to calibrate my models on a new dataset, I'm not sure I would want to switch to this new method. The interpretations seem rather difficult and not clearly beneficial. I appreciate how the authors are quite upfront about these shortcomings. There are some enlightening small pieces worthy of a wider audience in the paper; for example, how temperature scaling fails at multi-class calibration (figure 1). The regularization method also seems clearly beneficial. Overall, I think this paper is borderline, interesting but somewhat incremental.

Reviewer 4



POST REBUTTAL COMMENTS This did not affect my decision and score, but the citations on calibration should be improved. The oldest work on calibration that the authors cite is from 2000 (Platt). The definitions of calibration/calibration error, and many of the key ideas were proposed several decades ago by statisticians and meteorologists like Brier, Murphy, Winkler, Deegroot, Fienberg. The calibration error metric proposed there is different from ECE (root-mean-squared instead of mean absolute value), it would be good if the authors can mention that the RMS calibration error is another possible metric. I've included some pointers to the literature below (there are many other papers, but it should be OK to just cite a few): - Verification of forecasts expressed in terms of probability. GW Brier 1950. - Reliability of subjective probability forecasts of precipitation. AH Murphy and RL Winkler 1977. - The comparison and evaluation of forecasters. MH DeGroot and SE Fienberg 1983. Additionally, vector scaling is not neural specific, and the labeling of this should be changed (the labeling did not affect my decision either). I've read all the other reviews, the author response, and discussed the paper extensively with the other reviewers. I think this is a good piece of work, although there are several ways it can be strengthened. I stand by my original decision, that this paper is marginally above the acceptance threshold. I think the connection between Dirichlet calibration and Matrix calibration is neat. I do think it's a fairly natural extension of beta calibration: see page 4 of http://proceedings.mlr.press/v54/kull17a/kull17a.pdf where those authors assume P(model outputs | class) is a Beta/2-variable-Dirichlet distribution for each class, and derive the connection to Platt scaling. Here, this paper assumes P(model outputs | class) is a k-variable-Dirichlet (k > 2), and derive the connection to matrix scaling. However, it is still a good piece of work, and to the best of my knowledge nobody seems to have discovered this extension in 2 years. Detailed thoughts on author response: - Similarity between matrix scaling and Dirichlet scaling: In their response, the authors say they are not aware of using log-transformed class probabilities in multi-class calibration, so composing log-transform with matrix scaling is novel. This does not convince me. In fact, regular vector scaling is done *before* the softmax, which is very close to log-transformed probabilities. Additionally, as I understand using log-transformed probabilities is standard when doing calibration on probabilities (it generally works better). Further, note that vector scaling can essentially be viewed as their method with a very high regularization penalty. - Generalizing to other exponential families: The authors say the significance of their theoretical results is they can extend it to other exponential families. In this context, isn't the Dirichlet the most general member in the discrete exponential family? That is, if the set of possible outcomes is {x \in R^k | x_i \in {0, 1}, sum x_i = 1}, as it is in their multiclass setting, then the most general distribution on x is a k-category Dirichlet distribution. So maybe the authors are suggesting they could add constraints, perhaps if some classes seem like they could be more related than others. To strengthen the paper, the authors could sketch this out in more detail. - Neural effect sizes: for the neural experiments, the authors should explain why the improvement in log-loss/Brier score (e.g. 0.8%/0.7%) is significant (even if classwise-ECE drops by 2%). I don't know if this is very positive or not. - Non-neural experiments: I'm less familiar with non-neural calibration. My reservations are: they did not try vector scaling on non-neural calibration (after applying log-transform), even though this method does best in the neural experiments and would be the natural baseline. I've discussed with the other reviewers, and we all agree that vector scaling is not "neural specific". Isotonic regression does better in classwise-ECE (but not statistically significant), and 1% worse in Brier score (not sure if this is statistically significant). Isotonic is going to be worse in log loss, because the authors' implementation uses sklearn which optimizes MSE, and further the probabilities are then normalized, but the normalized version is not trained to optimize MSE/log-loss. - In general, I agree with R2 and R3 that it's not convincing that we should use the new methods over vector scaling. But a calibration method does not need to be better in *all* datasets. It is quite likely that each method is more amenable to certain kinds of data. As mentioned in the review, they could show that there is a significant difference in certain kinds of datasets. - The p-classwise-ECE metric can be better motivated (e.g. looking at table 20, in many cases all methods get a score of 0/are considered not calibrated). Their argument for non-neural experiments is that p-classwise-ECE and classwise-ECE show complementary information, but p-classwise-ECE is significant while classwise-ECE is not, is that a good enough reason for using the former? My take is that if the p-classwise-ECE is better, but classwise-ECE is worse, perhaps that suggests that there are a small number of datasets where the method does much worse, but it often does slightly better? That could still be useful, but I'm not 100% clear about what's going on here. ----------------------------------------- Originality: Medium. - The authors say that they demonstrate that the multiclass setting introduces numerous subtleties that have not been recognized by other authors. Nixon et al 2019 point out similar issues, and their metric is the same as classwise ECE. They also have some experiments showing that temperature scaling performs worse than vector scaling in this metric. This paper provides a lot more extensive evidence of this, but it does reduce the novelty of this paper. - They give a probabilistic interpretation for matrix scaling, showing an equivalence to assuming P(q | Y) ~ Dirichlet(alpha). There is some novelty here. However, it is important to note that the insights are similar to Kull, Filho, and Flach 2017. In that paper on beta calibration, those authors show an equivalence between assuming P(q | Y) ~ Beta(alpha) and (a minor variant of) Platt scaling. This paper’s result is basically the multi-dimensional extension of that result. This is not a bad thing - it just makes me borderline on novelty here. The authors should comment more on the relative novelty compared to that paper - what are some differences in the proof techniques, and what potential impact might this have? - The off diagonal regularization for matrix scaling is novel to the best of my knowledge, although it is worth noting that the prior method (vector scaling) is an extreme version of this, where all the off diagonal elements are set to 0. - Related work is well covered. The paper could cite the Hosmer Lemeshov test (Hosmer and Lemeshov 1980), which is a significance test for miscalibration, which appears close to some of the metrics they use. They could also cite Kuleshov and Liang 2015 which deals with calibrated structured prediction, but a specific instantiation of their method is for multi-class calibration. They have a notion of “event pooling” and their metric is weaker than classwise calibration. They could cite Nixon et al. Quality: Medium-High - The experiments are extensive and well done, and I commend the authors for including so many details (on many datasets and metrics). These details could guide practitioners in the future. That said, I have a few suggestions and questions. - If we look at classwise-ECE (table 10 supplementary), or confidence-ECE (table 9) isotonic regression seems to do better. Why is p-classwise-ECE chosen over classwise-ECE in the main paper? It’s also hard for me to ascertain the effect size of many of these results - significance and effect sizes can be very different things. For example, their results show that dirichlet scaling typically leads to better accuracy (but worse classwise-ECE) than isotonic regression on the non-neural network experiments - but what are the effect sizes of these? For the neural network experiments the effect sizes look rather small, and vector scaling does better at classwise-ECE. The paper says that the effect size is small (2%), and regularized matrix scaling does better at log loss, but the latter effect size is very small too. Given the similarity of these methods, I’m not convinced about the value of their regularized matrix scaling. - Minor: for the non-neural network experiments, it wasn’t clear to me which methods were better at calibrating which models. One could imagine that all methods are roughly the same at calibrating SVMs, MLPs, but some methods are much better at calibrating decision trees and Naive Bayes (which might have very different kinds of biases). I think this is a minor issue, but the authors could think about whether there is a simple way to show this. Again, I realize the authors provide ranks in the supplementary material e.g. table 10, but effect sizes (for at least some of these) would be good. Clarity: Medium-High. The paper is well written. There are some typos, for example the average ranks in table 3 and 4 are identical. The section on “Interpretability” was confusing - I didn’t really follow that paragraph and the associated Figure 2. It also isn’t clear why vector scaling isn’t a general-purpose calibrator - for methods that output “probabilities” (like Naive Bayes), can’t you apply a log transform and then do vector scaling? Significance: Medium. I think this paper has interesting content. However, given that prior work has already focused on classwise ECE, and their proposed method does not appear to do better than prior methods (isotonic regression, vector scaling), I’m borderline on the significance. The connection between Dirichlet calibration and matrix scaling could lead to future ideas, and their focus on multiclass calibration is in the right direction (they have far more extensive experiments than Nixon et al, and the latter paper only came out a couple of months before NeurIPS deadline) which could be useful for the community. References: Measuring Calibration in Deep Learning. Jeremy Nixon, Michael Dusenberry, Linchuan Zhang, Ghassen Jerfel, Dustin Tran. Arxiv 2019. Beta calibration: a well-founded and easily implemented improvement on logistic calibration for binary classifiers. Meelis Kull, Telmo de Menezes e Silva Filho, Peter Flach. AISTATS 2017. Calibrated Structured Prediction. Volodymyr Kuleshov and Percy Liang. NeurIPS 2015. Goodness of fit tests for the multiple logistic regression model. Hosmer, D.W., Hosmer, T. and Lemeshow, S. Communications in Statistics - Theory and Methods 1980.

[Author Response · NeurIPS 2019]

We thank the reviewers for their very helpful feedback.

**R1,R3,R4:** Interpretability and Fig.2. **Answer:** Our text about interpretability was not clear enough and we will add
the following: "In the canonical parametrisation, increasing the value of element $a_{ij}$ in matrix **A** increases the calibrated
probability of class $i$ (and decreases the probabilities of all other classes), with the amount of change depending on the
uncalibrated probability of class $j$. E.g., element $a_{2,8} = 0.63$ of Fig.2b increases class 2 probability whenever class 8
has high predicted probability, modifying decision boundaries and resulting in 26 less confusions of class 2 for 8 as seen
in Fig.2c. Looking at the matrix **A** and vector **c**, it is hard to know the effect of the calibration map without performing
the computations. However, at $k+1$ *'interpretation points'* this is (approximately) possible. One of these is the centre
of the probability simplex, which maps to **c**. The other $k$ points are vectors where one value is (almost) zero and the
other values are equal, summing up to 1. Fig.2a shows the 3+1 interpretation points in an example for $k = 3$, where
each arrow visualises the result of calibration (end of arrow) at a particular point (beginning of arrow). The result of
calibration map at the interpretation points in the centres of sides (facets) is each determined by a single column of **A**."

**R4:** "...Dirichlet calibration is very slightly different from matrix scaling..." **Answer:** While Dirichlet calibration
can be viewed as a simple modification to matrix scaling (Dirichlet calibration is equivalent to the composition of
log-transforming class probabilities and applying matrix scaling), we believe it is a novel modification as we are not
aware of earlier works using log-transformed class probabilities in multi-class calibration (other than in the special case
of binary calibration with beta calibration).

**R4:** "...why this is not a straightforward extension of beta calibration?" **Answer:** The derivation's starting point for
Dirichlet calibration is indeed a straightforward generalization of the starting point of beta calibration [11]. Proof of our
Theorem 1 is technically much more involved than the proofs in [11], but intuitively follows the same logic. The new
challenge when moving from binary to multi-class was choosing the right regularization method (no regularization was
applied in [11] for beta calibration).

**R4:** "I like the connection between Dirichlet calibration and matrix scaling. I'm just not sold on its significance.
**Answer:** With this paper we have explored the family of calibration maps derived from Dirichlet distributions. The
approach is however more general: e.g., it opens up the possibility of deriving further calibration maps from other
members of the exponential family.

**R4:** "Explain why your method is significant even though it appears to have lower classwise-ECE than isotonic regres-
sion/vector scaling." **Answer:** Good calibration performance is easier to achieve with worse predictive performance
(see also lines 108-113 of the main paper). Our methods improve predictive performance (e.g. log-loss) over the
state-of-the-art, while retaining a comparable level of calibration. Based on critical difference diagrams, there are no
significant differences in classwise-ECE between OvR_Isotonic and our Dirichlet_L2 (Supp.Fig.6f, non-neural) and
between VecS and our MS-ODIR (Supp.Fig.11f, neural). Our methods have non-significantly worse classwise-ECE but
better p-classwise-ECE (Supp.Figs.6h,11h).

**R4:** "Why is p-classwise-ECE chosen over classwise-ECE in the main paper?" **Answer:** These measures provide
complementary information but due to space constraints we omitted classwise-ECE which does not show significant
differences and we show it only in the supplementary.

**R2:** "This raises a general question of whether modeling pairwise class interactions is necessary or beneficial for
multi-class calibration." **Answer:** MS-ODIR has lower log-loss than VecS in 13 out of 14 cases (Table 4) and this can
only be due to non-zero off-diagonal values, indicating the importance of modelling pairwise class interactions. We
apologize for the copy-paste error of average ranks in Table 4 (thanks to R2&R4 for noticing), the correct average ranks
are $6.0, 3.5, 3.79, 2.93, 3.14, 1.64$ (as correctly shown in Supp.Tab.13).

**R4:** "...what are the effect sizes...?". **Answer:** In neural experiments, the effect size in predictive performance
improvement of MS-ODIR over VecS measured by median relative reduction of loss (i.e. $(VecS - MS\_ODIR)/VecS$))
is 0.5% for error rate, 0.8% for log-loss and 0.7% for Brier score (calculated from Supp.Tabs.18,13,14). In non-neural
experiments, the median relative loss reduction of Dir_L2 over OvR_Isotonic (i.e. $(Iso - Dir)/Iso$)) is 1.2% for error
rate, 13.8% for log-loss and 1.0% for Brier score.

**R3:** "...I'm not sure I would want to switch to this new method." **Answer:** We suggest to switch to Dirichlet_L2
(non-neural) or MS-ODIR (neural), as they improve predictive performance and are on par with the state-of-the-art in
calibration. These models are unique in modelling directly pairwise class interactions, and are likely to be particularly
valuable when recalibrating after any dataset shift.

We thank **R4** for the suggestion to study which methods are better at calibrating which models (remains as future work)
and for the suggested articles. We will cite Nixon et al. (6 weeks before us, independently, used SCE metric equivalent
to our classwise-ECE), Kull et al. (earlier version of beta calibration [11]), Kuleshov and Liang (used one-vs-rest
multi-class calibration), Hosmer and Lemeshov (studied model fitness in general, which partially inspects calibration).

[Meta-Review · NeurIPS 2019]

The paper completes the picture of post-training calibration by proposing Dirichlet calibration as a natural generalization of Beta calibration to the multi-class setting, and showing the connection between it and matrix scaling in the context of neural net models. The comprehensive experiments with both deep neural nets and non-neural models comparing a variety of post-training calibration techniques are also a strong point of the paper and was appreciated by all reviewers. On the negative side, the results are mixed with performance differences between the new techniques and other approaches being rather small. The authors should incorporate the reviewers' comments (R4 gave very detailed and thoughtful post-rebuttal comments). In particular the authors should: - cite older calibration work from statistics (see R4 comments for references). - experiment with vector scaling for non-neural methods. - analyze the whether the ODIR is so strong that the matrix scaling reverts to vector scaling, which would explain the marginal improvements over vector scaling - maybe discuss in more detail the use of other exponential families in place of Dirichlet.